# 53BP1-RIF1 and DNA-PKcs show distinct genetic interactions with diverse chromosomal break repair outcomes

Kaela Makins [1,2], Metztli Cisneros-Aguirre[1,2,3], Felicia Wednesday Lopezcolorado[1] & Jeremy M. Stark [1,2] ✉

53BP1 accumulates at DNA double strand breaks (DSBs) and is implicated in non-homologous end joining (NHEJ), but the genetic interplay of 53BP1 with the NHEJ pathway (e.g., DNA-PKcs) is poorly understood. We examine blunt DSB NHEJ of Cas9 DSBs, which is dependent on core NHEJ factors, and find that loss of 53BP1 does not affect such repair but causes a reduction when combined with DNA-PKcs disruption. In contrast, disrupting 53BP1 and DNA-PKcs, alone and together, has similar effects on the type of deletion mutation (increase in microhomology deletions). We find similar effects with RIF1, such that 53BP1-RIF1 appear to play a backup role for DNA-PKcs during blunt DSB NHEJ, but function in the same pathway to suppress microhomology deletions. Finally, DNA-PKcs kinase inhibition causes increased radiosensitivity and homology-directed repair that is not additive with loss of 53BP1. Altogether, 53BP1-RIF1 and DNA-PKcs show distinct genetic interactions with diverse DSB repair outcomes.

Chromosomal DNA double-strand breaks (DSBs) can be caused by physiological processes, as well as clastogenic cancer therapeutics (e.g., radiotherapy), or with targeted nucleases for gene editing. DSBs can be repaired by a number of pathways[1,2]. For one, homology-directed repair (HDR) uses invasion and replication of a template (e.g., the sister chromatid) to bridge the DSB. In contrast, non-homologous end joining (NHEJ) repair involves ligation of the DSB ends without invasion of a homologous template. There are several sub-pathways of chromosomal DSB end joining repair that differ by the factors involved, the possible substrates for ligation, and genetic outcomes. To avoid confusion between total DSB end joining repair and the canonical NHEJ pathway, we refer to all DSB end joining simply as EJ. The NHEJ pathway involves DNA-PKcs, Ku70/Ku80, XRCC4, and XLF, which synapse DSB ends, regulate DSB end processing, and position DNA Ligase 4 (LIG4) to catalyze the ligation[3–8]. NHEJ is critical for DSB repair events that are not stabilized by an annealing intermediate (e.g., blunt DSB ends), as well as EJ events involved in antibody maturation (i.e., V(D)J and Class Switch Recombination)[3–6,9,10]. In the absence of

NHEJ factors, there are backup EJ pathways that function to varying degrees based on the specific context (e.g., DSB end structure, organism, and cell type)[11–16]. Such backup EJ pathways appear to cause repair outcomes with greater microhomology at the junctions, indicating that microhomology annealing is involved in bridging the DSBs[11–15]. Such annealing appears to be mediated by DNA polymerase theta (POLθ) and such repair is referred to as theta-mediated EJ (TMEJ), although DNA polymerase lambda also appears to play a role[17,18].

53BP1 is a key DNA damage response factor that is implicated in NHEJ[19–21], but its precise role in DSB repair outcomes, particularly in relation to core NHEJ factors, has remained unclear. 53BP1 was initially identified in a two-hybrid screen for interacting proteins with the p53 tumor suppressor[19–21]. Subsequent characterization found that 53BP1 can form robust foci at DSBs in cells, and can bind DNA and enhance DNA ligase 4 activity[19–21]. Additional evidence that 53BP1 is involved in DSB repair, and NHEJ in particular, is its key role in class switch recombination, and fusion of deprotected telomere ends[22,23]. However, 53BP1 has relatively modest effects on V(D)J recombination, which is

[1]Department of Cancer Genetics and Epigenetics, Beckman Research Institute of City of Hope, Duarte, CA, USA. [2]Irell and Manella Graduate School of Biological Sciences, Beckman Research Institute of the City of Hope, Duarte, CA, USA. [3]Present address: Salk Institute for Biological Sciences, La Jolla, CA, USA. ✉e-mail: jstark@coh.org

dependent on core NHEJ factors (e.g., XRCC4)[24,25]. Another key distinction between 53BP1 and NHEJ is their influence on homology-directed repair (HDR). Namely, 53BP1 is critical to suppress HDR in BRCA1-deficient cells, i.e., loss of 53BP1 rescues HDR in BRCA1-deficient cells[26,27]. In contrast, loss of NHEJ factors (e.g., LIG4) does not

appear to cause such rescue[26]. To suppress such HDR, 53BP1 appears to block DSB end resection in BRCA1-deficient cells[26,27].

There are several other distinctions between 53BP1 and core NHEJ factors. For example, whereas the NHEJ complex is centered around the KU heterodimer binding DNA ends[3–7], 53BP1 appears to associate with DSBs by binding to specific chromatin marks adjacent to DSBs[28]. Such chromatin binding by 53BP1 promotes DSB recruitment of RIF1, a factor that binds phosphorylated 53BP1[28–39]. In vertebrates, RIF1 is critical for the timing regulation of DNA replication initiation and is a mediator for 53BP1-dependent DSB repair[28–40]. Regarding the latter function, RIF1 bridges the REV7-Sheildin-CST complex to 53BP1-bound chromatin at DSB sites, which is important to suppress DSB end resection[28–39]. In addition to suppressing end resection via Shieldin, 53BP1 has also been implicated in DSB end tethering and mobility at long distances[22–25], as well as compaction of chromatin at DSBs[41]. These latter functions of 53BP1 may be linked to its capacity to oligomerize and form large condensates[42], as well as form microdomains along with RIF1[41]. Finally, 53BP1 has other effector proteins apart from RIF1-Shieldin that could influence DSB repair, including CTC1–STN1–TEN1, ASTE1, PAXIP1/PTIP, DYNNL1, and TP53[28,35,37,43–45]. In summary, 53BP1 is distinct from the NHEJ pathway, and its interplay is poorly understood. Thus, we sought to examine the genetic relationships between 53BP1-RIF1 and the NHEJ pathway on diverse DSB repair outcomes, focusing on the DNA-PKcs kinase subunit of NHEJ.

## Results

### Combining loss of 53BP1 with an inhibitor of DNA-PKcs causes a decrease in No Indel EJ

We have sought to define the influence of 53BP1 on chromosomal break end joining (EJ), including whether this factor is partially redundant with NHEJ factors. To begin with, we examined the role of 53BP1 in blunt DSB EJ, which is dependent on several NHEJ factors[9]. Specifically, we used HEK293 cells with the chromosomal EJ7-GFP reporter, which measures EJ between two Cas9-induced blunt DSBs that are ligated without inserted or deleted nucleotides (nt) (No Indel EJ, Fig. 1A)[9]. This No Indel EJ repair outcome restores a codon that is essential for GFP function (Gly67)[46], causing GFP+ expression in cells, which can be measured via flow cytometry[9]. The structure of the GFP+ repair product has also been confirmed by GFP cell sorting and sequencing[9]. This repair event requires the NHEJ factors XRCC4 and KU, is largely dependent on XLF, and is only partially dependent on DNA-PKcs[47].

We considered that 53BP1 may be partially redundant with DNA-PKcs for No Indel EJ. We based this hypothesis on evidence that both of these factors are implicated in DSB end synapsis and regulation of DSB end processing[3–8,22–27,41]. Using the EJ7-GFP reporter integrated into HEK293 cells (Parental), we generated a 53BP1 knockout (53BP1-KO) cell line, and examined the frequency of this repair event, both with and without treatment with a DNA-PKcs kinase inhibitor (M3814, 500 nM)[48,49]. We also included transient expression of 53BP1 to test for complementation. We found no significant difference in No Indel EJ frequencies between Parental and 53BP1-KO cells, and expression of 53BP1 in the 53BP1-KO caused a modest increase (1.4-fold, Fig. 1B, C). These findings are similar to prior studies with 53BP1-KO mouse embryonic stem cells (mESCs)[50]. However, M3814 treatment caused a reduction in No Indel EJ that was further reduced with 53BP1 loss (1.8-fold), which was complemented with expression of 53BP1 (3-fold). Namely, the fold effect of M3814 was greater in 53BP1-KO cells vs. the parental cell line and the 53BP1-KO complemented with 53BP1 (Supplementary Fig. 1). As a control, we found that 53BP1 loss does not affect M3814 inhibition of DNA-PKcs kinase activity (Supplementary Fig. 2). These findings indicate that loss of 53BP1 magnifies the effect of DNA-PKcs inhibition on No Indel EJ.

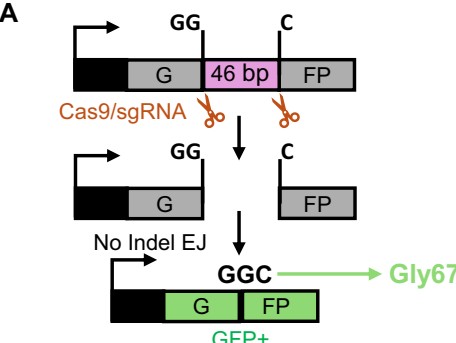

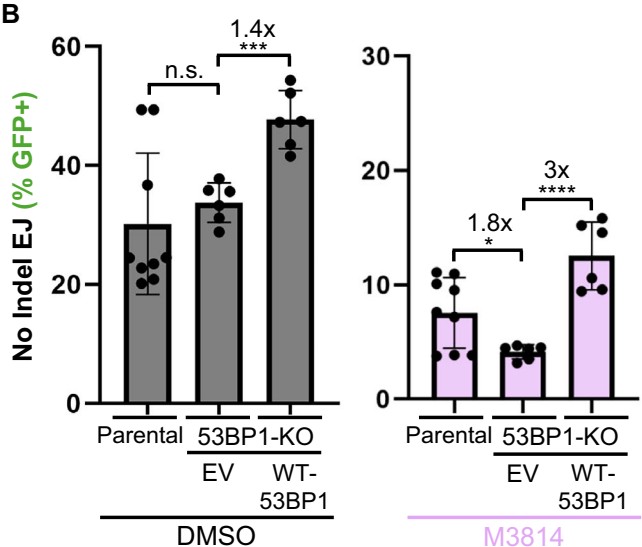

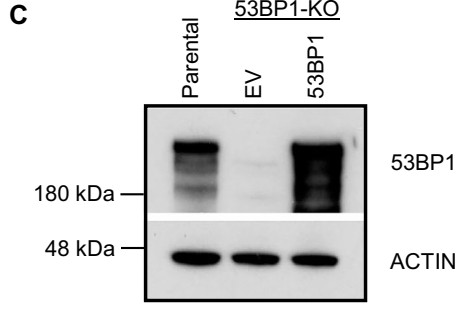

**Fig. 1 | Combining the loss of 53BP1 with an inhibitor of DNA-PKcs causes a decrease in No Indel EJ. A** Schematic of the EJ7-GFP reporter assay, which is chromosomally integrated in HEK293 cells and measures No Indel EJ (i.e., repair of blunt DSB distal ends of two Cas9 DSBs, which restores the critical Gly67 codon of GFP). Cells are transfected with Cas9 and two sgRNA expression vectors, and GFP frequencies are normalized to transfection efficiency with parallel GFP transfections. **B** Loss of 53BP1 magnifies the effect of DNA-PKcs inhibition (M3814) on No Indel EJ. Frequencies represent the mean ± SD. $n = 6$ biologically independent transfections, except Parental cells $n = 9$. Statistics with an unpaired $t$-test using Holm−Sidak correction. *$P = 0.0203$, ***P = 0.0002, ****$P < 0.0001$, n.s. = not significant. **C** Immunoblot analysis of 53BP1 for the cell lines and conditions shown in (**B**). Source data are provided as a Source data file.

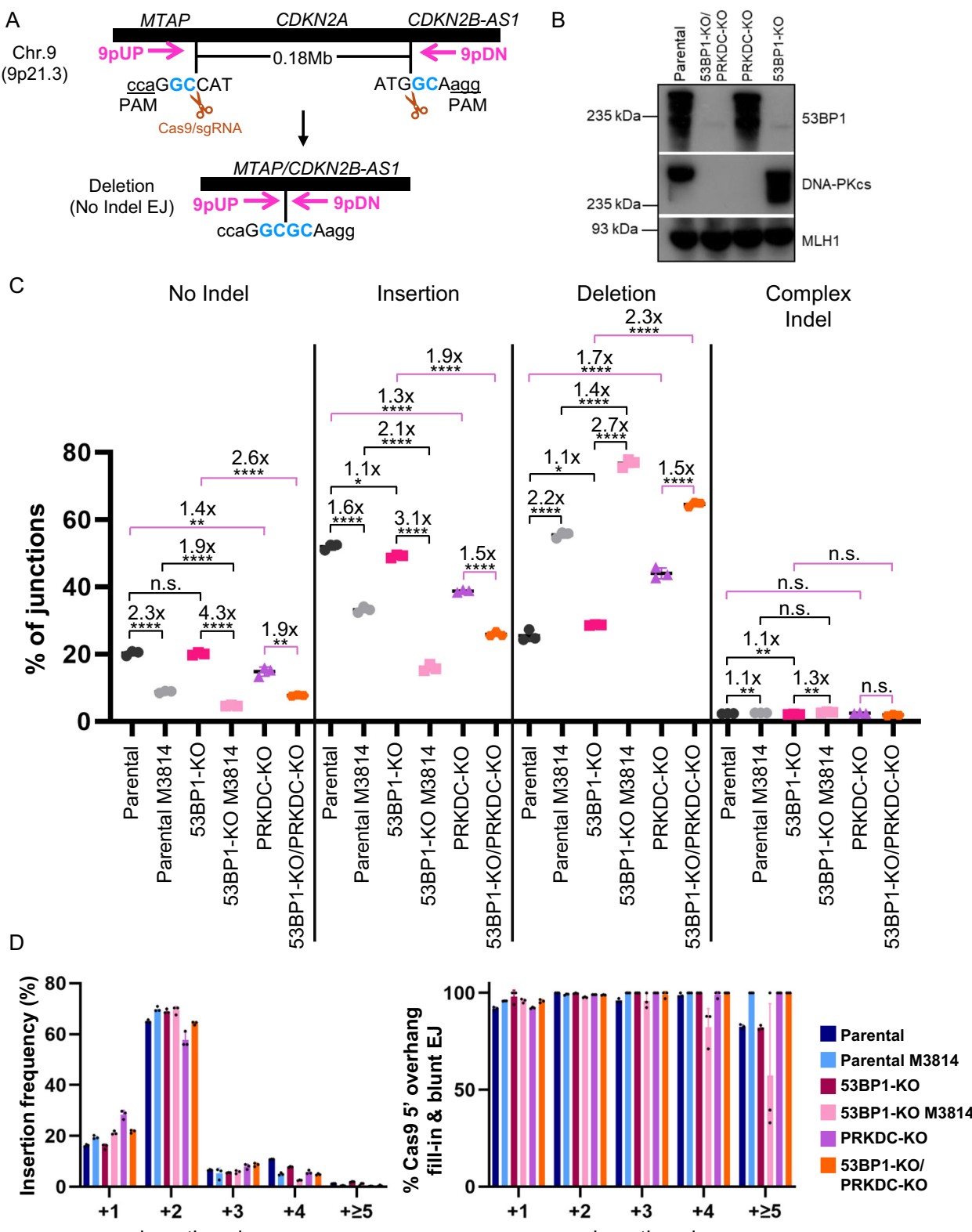

**Combining loss of 53BP1 with either a DNA-PKcs inhibitor or genetic loss causes a decrease in DSB blunt EJ**

We then sought to use a distinct assay to examine the interplay between 53BP1 and DNA-PKcs on diverse EJ outcomes. Also for this analysis, we sought to disrupt DNA-PKcs with both M3814 treatment as well as genetic loss of DNA-PKcs (PRKDC-KO). For this, we used an MTAP/CDKN2B-AS1 deletion (MA-del) assay to examine various EJ outcomes via the chromosomal deletion rearrangement of the CDKN2A gene, a common deletion rearrangement found in various cancer types (Fig. 2A)[51,52]. Expression of Cas9 and two sgRNAs induces two DSBs that target two loci that flank the CDKN2A gene (MTAP and CDKN2B-AS1, Supplementary Table 1). Subsequently, the deletion rearrangement is amplified using PCR, and amplicons are analyzed using deep sequencing. The reads are then aligned to the No Indel EJ product, i.e., repair of the distal DSB blunt ends without Insertions or Deletions. The analyzed reads are then categorized as No Indel EJ,

**Fig. 2 | Combining loss of 53BP1 with either a DNA-PKcs inhibitor or genetic loss causes a decrease in DSB blunt EJ. A** Schematic of the MTAP/CDK2NB1-AS1 deletion rearrangement reporter (MA-del), which induces two Cas9 DSBs that flank the *CDKN2A* locus. PCR is used to amplify the deletion rearrangement in preparation for deep sequencing analysis. **B** Immunoblot analysis of 53BP1 and DNA-PKcs in the Parental, single KO, and double KO cell lines. Immunoblot analysis of the genotype of the cell lines was performed at least twice. **C** Loss of 53BP1 alone has no obvious effect on indel types, but when combined with DNA-PKcs inhibition or loss, 53BP1 loss causes a decrease in No Indel EJ and Insertions, and an increase in Deletions. Shown are junction frequencies for No Indel EJ, Insertions, Deletions, and

Complex Indels for each condition. Frequencies represent the mean ± SD. $n = 3$ biologically independent transfections. Statistics with an unpaired $t$-test using Holm−Sidak correction. *left to right $P < 0.003615$, $P = 0.027318$, **left to right $P = 0.003615$, $P = 0.000872$, $P = 0.006465$, $P = 0.006120$, $P = 0.002870$, ****$P > 0.0001$, n.s. = not significant. **D** Insertion events are likely associated with Cas9 5′ overhang fill-in and blunt DNA EJ in all conditions. Shown is the frequency of Insertion sizes and percent Insertions consistent with Cas9 5′ overhang fill-in and blunt EJ for the experimental conditions shown in (**C**). Frequencies represent the mean ± SD. $n = 3$ biologically independent transfections. Source data are provided as a Source data file.

---

Insertion, Deletion, and Complex Indel repair outcomes (Supplementary Table 2).

We tested this assay on both Parental and 53BP1-KO cells with and without M3814 treatment, PRKDC-KO cells, as well as double knockout 53BP1-KO/PRKDC-KO cells (Fig. 2B). Beginning with DNA-PKcs disruption, we found that M3814 treatment and DNA-PKcs loss each caused a decrease in No Indel EJ and Insertions, and an increase in Deletions (Fig. 2C), which is consistent with a recent report[53]. In contrast, 53BP1 loss alone had no obvious effect on the relative frequencies of these EJ categories. However, in cells treated with M3814, loss of 53BP1 caused a decrease in No Indel EJ (1.9-fold) and Insertions (2.1-fold), and an increase in Deletions (1.4-fold) (Fig. 2C). Similarly, with PRKDC-KO cells, loss of 53BP1 caused a decrease in No Indel EJ (1.9-fold) and Insertions (1.5-fold), and an increase in Deletions (1.5-fold) (i.e., comparing 53BP1-KO/PRKDC-KO vs. PRKDC-KO, Fig. 2C). Finally, in all conditions, Complex Indels were rare (Fig. 2C). These findings indicate that loss of 53BP1 alone does not cause an obvious effect on these EJ categories, but when combined with DNA-PKcs disruption via M3814 treatment or PRKDC-KO, causes a significant decrease in No Indel EJ and Insertion frequencies, and a significant increase in Deletion frequencies.

Because frequency patterns were similar between No Indel EJ and Insertion outcomes, we hypothesized that Insertion outcomes are caused by blunt DSB EJ. Specifically, we posited that Insertion mutations are caused by staggered Cas9 DSBs that cause 5′ overhangs that are filled in to generate blunt DSBs prior to EJ, as has been shown in several studies[10,53,54]. Insertions caused by this mechanism would result in specific sequences. Thus, we analyzed the Insertion sequences with the MA-del assay, and found that the vast majority of Insertions are consistent with this mechanism (Fig. 2D, Supplementary Fig. 3), as found in a recent report[53]. For example, +2 nucleotide Insertions were the most frequent Insertion size and nearly all of the sequences consistent with staggered Cas9 DSBs that are filled in prior to blunt EJ, in all cellular conditions (Fig. 2D). Thus, the Insertions appear to result from blunt EJ events, as are the No Indel EJ events. Accordingly, our findings indicate that 53BP1 is dispensable for blunt EJ unless combined with loss or inhibition of DNA-PKcs. We suggest that 53BP1 promotes DSB end synapsis during EJ, but this function appears to be dispensable in the presence of DNA-PKcs.

## 53BP1 loss and DNA-PKcs disruption, alone and together, cause a similar shift in deletion patterns

Apart from blunt DSB EJ per se, we then assessed whether 53BP1 affects the pattern of deletion mutations, with or without disruption of DNA-PKcs. Namely, using the deletion mutations identified in the MA-del experiments above, we tested whether 53BP1 affects deletion size and/ or microhomology use. Specifically, we analyzed the deletions to determine the frequency of deletions and the fraction of microhomology used for each deletion size (Fig. 3A, B, Supplementary Table 3). From this analysis, we found that loss of 53BP1 alone caused a striking change in the Deletion pattern: a significant increase in −2, −7, −75 deletion sizes, which are associated with ≥2 nt microhomology, and a significant decrease in several deletion sizes without

microhomology (i.e., 0, 1 nt) (Fig. 3A, B). There are a few notable exceptions to this pattern: loss of 53BP1 had no effect on the −13 deletion that is associated with microhomology, and interestingly caused a decrease in −17 and −22 deletions associated with microhomology. We then compared the effects of 53BP1 loss to M3814 treatment, and found a very similar pattern. Namely, M3814 treatment caused a significant increase in −2, −7, −75 deletions with microhomology, and a decrease in several deletion sizes that do not utilize microhomology, similar to a recent report[53] (Fig. 3A, B). Also, as with 53BP1 loss, M3184 treatment had no effect on the −13 deletion associated with microhomology. However, for the −17 and −22 deletions associated with microhomology, M3814 treatment did not cause a decrease in these events, as was caused by 53BP1 loss. Notably, M3814 treatment in 53BP1-KO cells had very little effect on deletion size, with the exception of causing an increase in the −17 deletion. Finally, the −20 deletion associated with microhomology was only modestly affected by 53BP1 loss and M3814 treatment, and other deletion sizes with or without microhomology from −23 to −85 nt were relatively rare in each of these conditions (Fig. 3A, B).

We then evaluated the genetic loss of DNA-PKcs. We found that such genetic loss (PRKDC-KO) caused a nearly identical effect on the deletion mutation pattern as M3814 treatment, and the combined loss of DNA-PKcs and 53BP1 (53BP1-KO/PRKDC-KO) was very similar to the loss of DNA-PKcs alone (Supplementary Fig. 4). Exceptions to this pattern were an increase in the −21 deletion without microhomology and a decrease in the −75 deletion with microhomology in the 53BP1-KO/PRKDC-KO vs. 53BP1-KO treated with M3814 (Fig. 3A, B, Supplementary Fig. 4).

## 53BP1 loss and DNA-PKcs disruption, alone and together, cause a similar increase in microhomology usage for deletions

The above analysis indicates that DNA-PKcs disruption and 53BP1 loss cause an increase in deletions with microhomology, and a reduction in deletions without microhomology. To examine microhomology another way, we compiled the microhomology usage for all deletions from the above MA-del experiment. From this analysis, 0–3 nt of microhomology were the most common in all conditions, but loss of 53BP1 and disruption of DNA-PKcs (M3814 and PRKDC-KO) caused a significant decrease in 0 and 1 nt microhomology deletions, and an increase in 2 nt and 3 nt microhomology deletions (Fig. 4). Furthermore, for these events, combined loss of 53BP1 with DNA-PKcs disruption was not different from the single knockouts. Deletions with 4 nt of microhomology were less common, but loss of 53BP1 and disruption of DNA-PKcs caused an increase in these events, except for PRKDC-KO/53BP1-KO cells. Altogether, these findings indicate that 53BP1 loss and DNA-PKcs disruption cause a similar increase in microhomology deletions, and that combined disruption of both factors is not additive. We suggest that DNA-PKcs and 53BP1 function in the same pathway to affect deletion size/microhomology patterns, which is a distinct genetic relationship we observed for the frequency of blunt DSB EJ (53BP1 appears to be a backup factor for DNA-PKcs). Accordingly, the genetic relationship between 53BP1 and DNA-PKcs is distinct for different aspects of DSB repair.

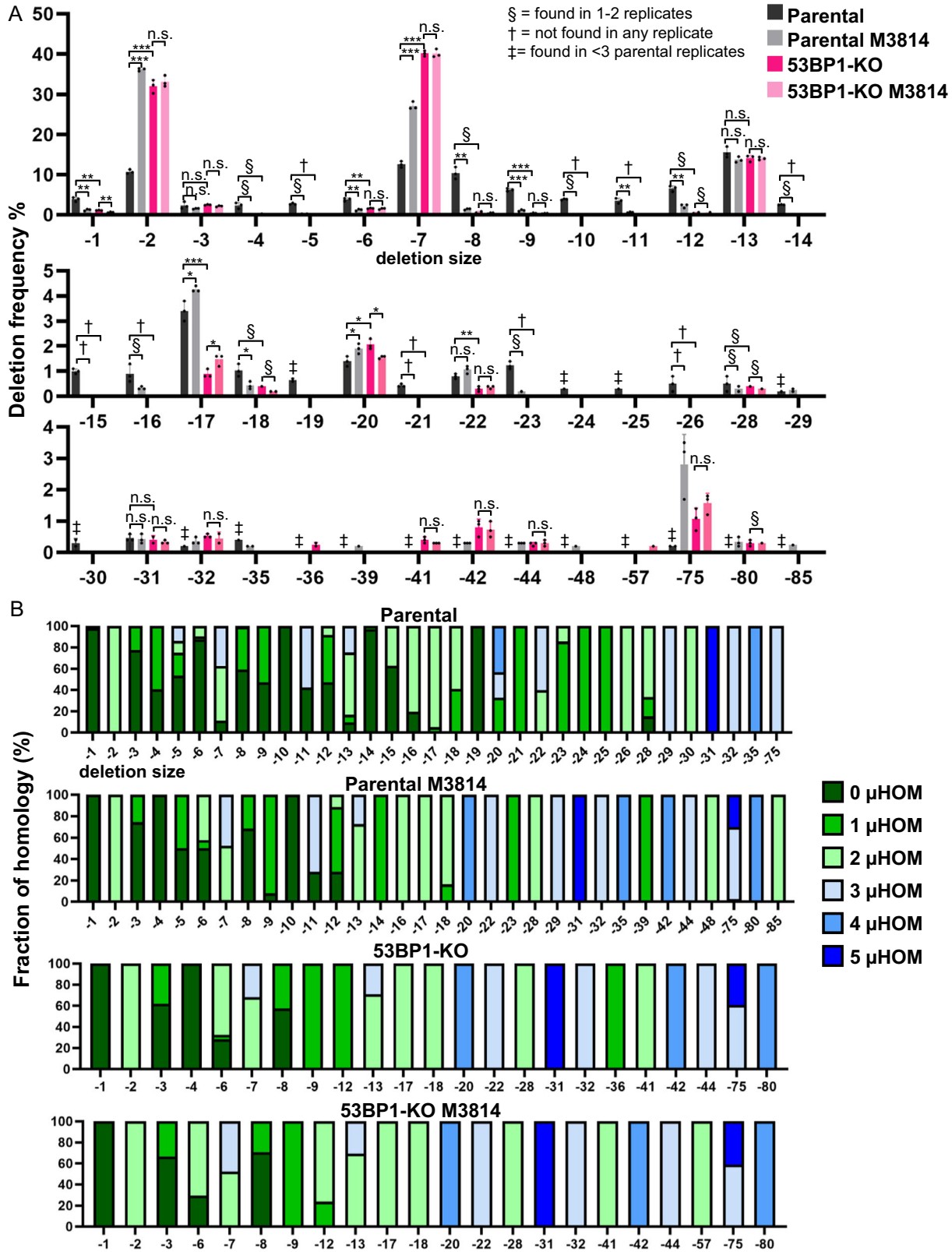

**Fig. 3 | 53BP1 loss and DNA-PKcs disruption, alone and together, cause a similar shift in deletion patterns. A** Loss of 53BP1 causes an increase in −2 and −7 deletions, similar to inhibition of DNA-PKcs, and DNA-PKcs inhibition does not cause a further increase in 53BP1-KO cells. Shown are deletion sizes for the samples shown in Fig. 2C. Frequencies represent the mean ± SD. $n = 3$ biologically independent transfections. Statistics with an unpaired $t$-test using Holm−Sidak correction. *$P < 0.05$, **$P < 0.01$, ***$P < 0.001$, ****$P < 0.0001$, n.s. = not significant. § = deletion size was only found in 1 or 2 replicates, † = deletion size was not found in any of the replicates, and ‡ = deletion size was found in <2 Parental replicates.

**B** Microhomology use varies with distinct deletions. Shown is the fraction of microhomology used for each deletion size and experimental condition shown in (Fig. 2C). $n = 3$ biologically independent transfections. Source data are provided as a Source data file.

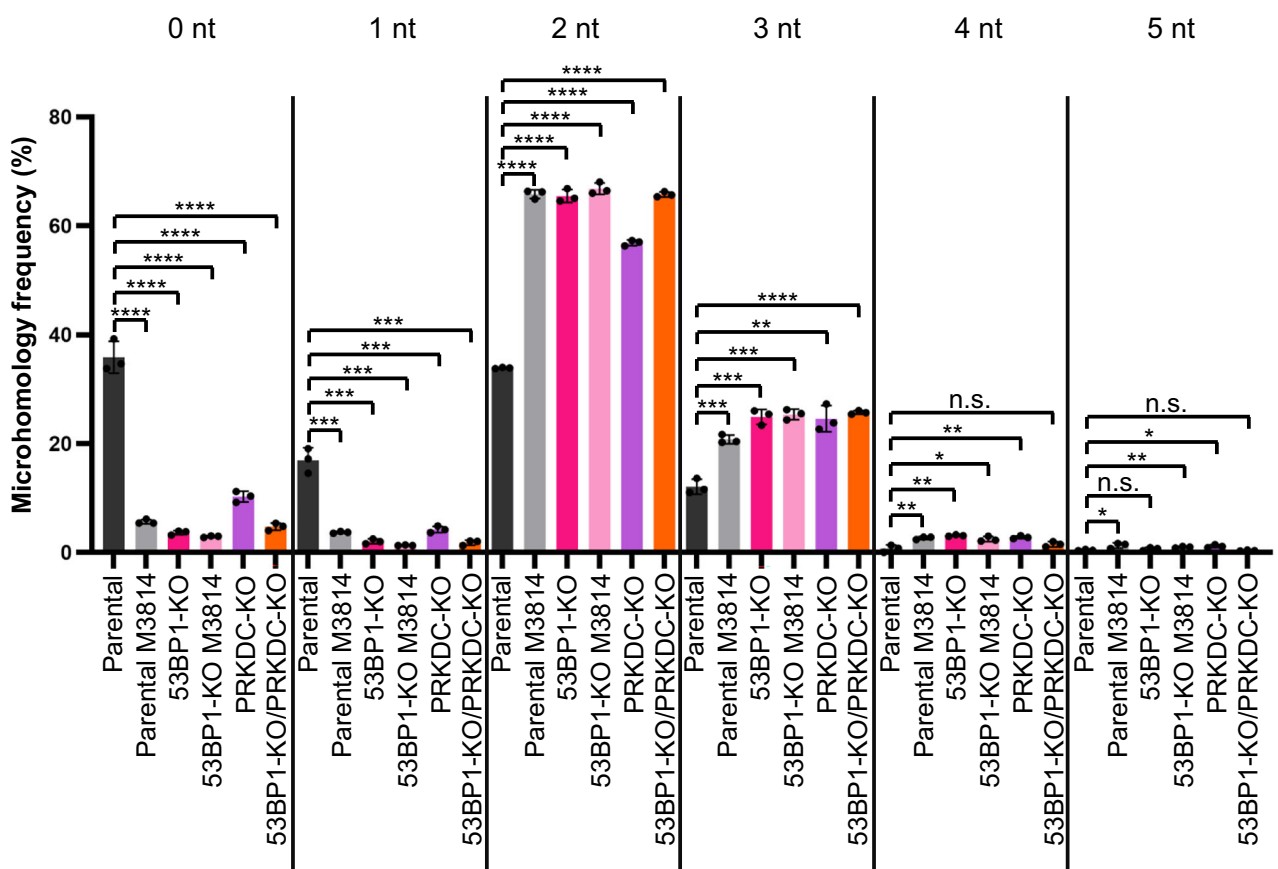

**Fig. 4 | 53BP1 loss and DNA-PKcs disruption, alone and together, cause a similar increase in microhomology usage for deletions.** Shown is the frequency of microhomology used amongst deletions for each experimental condition shown in (Fig. 2C). Frequencies represent the mean ± SD. *n* = 3 biologically independent transfections. Statistics with an unpaired *t*-test using Holm–Sidak correction. *left to right *P* = 0.021376, *P* = 0.044904, *P* = 0.013506, **left to right *P* = 0.001454, *P* = 0.009389, *P* = 0.003635, *P* = 0.007368, *P* = 0.006555, ***left to right *P* = 0.000556, *P* = 0.000556, *P* = 0.000289, *P* = 0.000728, *P* = 0.000363, *P* = 0.000671, *P* = 0.000332, *P* = 0.000158, ****P* < 0.0001, n.s. = not significant. Source data are provided as a Source data file.

## Loss of 53BP1 causes a decrease in DSB blunt EJ in XLF-deficient cells

We then considered that loss of the NHEJ factor XLF might also affect the relative influence of 53BP1 on EJ outcomes, because both XLF and DNA-PKcs are implicated in DSB end synapsis[6,55,56], and because loss of XLF revealed a role for 53BP1 in V(D)J recombination[24]. To begin with, we examined No Indel EJ using the EJ7-GFP reporter in XLF-KO and double mutant 53BP1-KO/XLF-KO cells (Fig. 5A, B, Supplementary Fig. 5). We found that XLF loss caused a marked decrease in No Indel EJ (7.9-fold), which was further reduced by 53BP1 loss (6.5-fold 53BP1-KO/XLF-KO vs. XLF-KO). Furthermore, transient expression of XLF and 53BP1 in these experiments restored No Indel EJ to the levels of Parental cells and XLF-KO cells, respectively (Fig. 5B, Supplementary Fig. 5). Thus, XLF is critical for No Indel EJ, as shown previously[9,57], yet the residual No Indel EJ in XLF-deficient cells is markedly dependent on 53BP1.

Next, we used the MA-Del assay to examine EJ outcomes in XLF-KO and double mutant 53BP1-KO/XLF-KO cells (Fig. 5C). From this analysis, we found that XLF loss causes a reduction in No Indel EJ and Insertions (2.4-fold and 3.4-fold, respectively), and an increase in deletions (3.6-fold), which is consistent with its role in blunt DSB EJ from other studies[9,53]. We also found that the frequency of No Indel EJ and deletions in the double mutant 53BP1-KO/XLF-KO was similar to the single XLF-KO, whereas Insertions were reduced 1.3-fold in the double mutant vs. XLF-KO (Fig. 5C).

We then examined Insertions in more detail. Beginning with Insertion size, the general pattern of Insertion frequencies remains consistent between the various cell lines, however the 53BP1-KO/XLF-KO showed a 1.3-fold decrease in 2 nt Insertions, and a 10.4-fold increase in ≥5 nt Insertions, compared to the Parental cells (Fig. 5D). Furthermore, the 53BP1-KO/XLF-KO showed a decrease in the frequency of insertions consistent with staggered Cas9 cleavage, followed by 5′ overhang fill-in and blunt EJ: i.e., a 1.4-fold decrease in 3 nt insertions, 2.3-fold decrease in 4 nt insertions, and a 27.3-fold decrease in ≥5 nt insertions (Fig. 5D). Thus, 53BP1-KO/XLF-KO showed a reduction in the frequency of Insertions (1.3-fold vs. XLF-KO), along with a decrease in the insertions consistent with blunt DSB EJ. Thus, these insertion results indicate that 53BP1 loss causes a decrease in blunt DSB EJ in XLF-KO cells. A caveat to this conclusion is that loss of 53BP1 did not affect the frequency of No Indel EJ in XLF-KO cells with the MA-del assay. We suggest that this discrepancy is due to the EJ7-GFP assay being more sensitive to changes in the efficiency of blunt DSB EJ. For example, XLF loss causes a much greater defect on No Indel EJ with the EJ7-GFP reporter vs. the MA-del assay. The reason for this distinction is unclear, but notably, the MA-del assay measures repair between two DSBs separated by a much larger distance vs. EJ7-GFP. In any case, altogether, these findings indicate that 53BP1 promotes DSB end synapsis during EJ, but this function is dispensable in cells with XLF and DNA-PKcs.

## Loss of 53BP1 and XLF causes distinct deletion patterns, and the effect of 53BP1 dominates in the double mutant

We then assessed the effect of XLF loss with and without 53BP1 loss may affect the pattern of deletion mutations (i.e., deletion size and

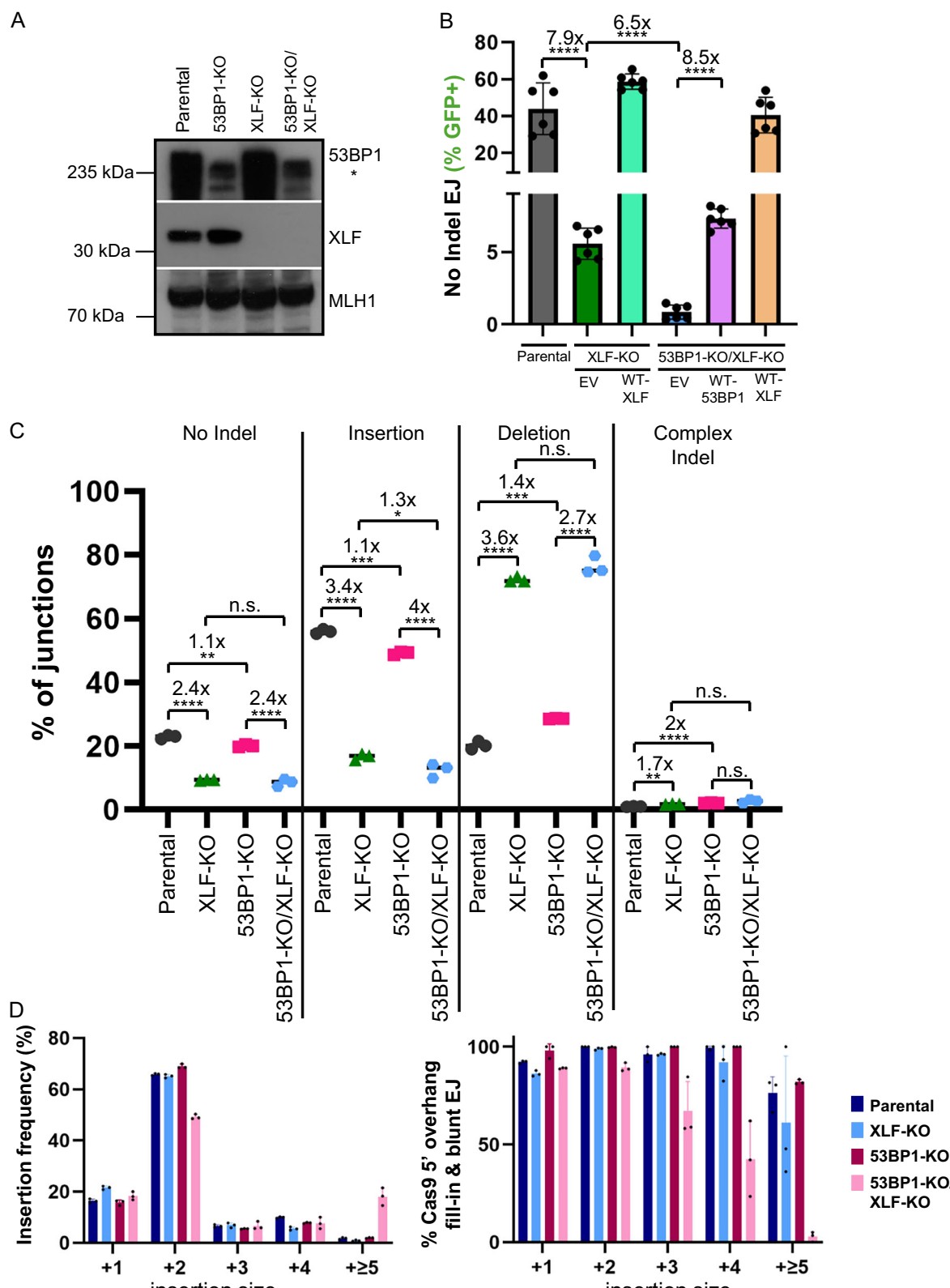

microhomology). From this analysis, we found that XLF loss caused a unique deletion pattern, similar to a recent study[53]. Specifically, loss of XLF did not have a substantial effect on the −2, −7, and −75 deletions associated with microhomology (Fig. 6A, B), whereas, as described above, loss of 53BP1 caused these events to increase (Fig. 3A, B). In contrast, loss of XLF caused an increase in several microhomology-associated deletions that range between −17 and −35 (Fig. 6A, B,

Supplementary Table 4). Finally, several deletion sizes without microhomology (i.e., 0−1 nt) were reduced by loss of XLF, which is similar to loss of 53BP1 (Fig. 6A, B). These findings indicate that while XLF loss and 53BP1 loss each cause an increase in deletions associated with microhomology, the deletion sizes are different. Consistent with this notion, when we analyzed microhomology usage independent of deletion size, both XLF loss and 53BP1 loss caused an increase in

**Fig. 5 | Loss of 53BP1 causes a decrease in DSB blunt EJ in XLF-deficient cells.**
**A** Immunoblot analysis of 53BP1 and XLF in the Parental, 53BP1-KO and XLF-KO, and double KO cell lines. **B** 53BP1 is critical for No Indel EJ in XLF-deficient cells, as measured with the EJ7-GFP reporter. Frequencies represent the mean ± SD. $n = 6$ biologically independent transfections, except Parental cells $n = 9$. Statistics with an unpaired t-test using Holm–Sidak correction. ****$P < 0.0001$, n.s. = not significant. **C** With the MA-del assay, combined loss of XLF and 53BP1 causes a reduction in Insertions, compared to XLF-KO. Shown are junction frequencies for No Indel EJ, Insertions, Deletions, and Complex Indels for each condition. Frequencies

represent the mean ± SD. $n = 3$ biologically independent transfections. Statistics with an unpaired t-test using Holm–Sidak correction. *$P = 0.037772$, **left to right $P = 0.002634$, $P = 0.002243$, ***left to right $P = 0.000164$, $P = 0.000369$, ****$P > 0.0001$, n.s. = not significant. **D** Insertion events are likely associated with Cas9 5′ overhang fill-in, and blunt DNA EJ are reduced in 53BP1-KO/XLF-KO cells. Shown is the frequency of Insertion sizes and percent Insertions consistent with Cas9 5′ overhang fill-in and blunt EJ for the experimental conditions shown in (**B**). Frequencies represent the mean ± SD. $n = 3$ biologically independent transfections. Source data are provided as a Source data file.

microhomology usage (e.g., reduction in 0–1, and increase in 2–3 nt, Supplementary Fig. 6).

Interestingly, the deletion pattern for cells with combined loss of 53BP1 and XLF largely mimics the deletion pattern of loss of 53BP1 alone. Namely, compared to Parental cells, the combined loss of 53BP1 and XLF caused a significant increase in −2 and −7 deletions associated with microhomology, similar to 53BP1-KO (Fig. 6A, B). Furthermore, the deletions associated with microhomology ranging between -17 and -35 that were substantially elevated in XLF-KO were similar between 53BP1-KO and 53BP1-KO/XLF-KO. As an exception to this pattern, the combined loss of 53BP1 and XLF failed to cause an increase in the −75 deletion size, which was similar to XLF-KO, not 53BP1-KO. Altogether, these findings indicate that loss of 53BP1 and XLF causes distinct deletion frequency patterns to each other, and the pattern caused by 53BP1 loss is largely dominant over XLF loss in the double knockout.

## Combined loss of RIF1 and DNA-PKcs inhibition causes a decrease in No Indel EJ

A downstream effector protein of 53BP1 is RIF1[28–39]. Thus, we considered that RIF1 and 53BP1 may have similar effects on EJ outcomes. For this, we generated a HEK293 RIF1-KO cell line with the EJ7-GFP reporter for No Indel EJ (Fig. 7A, B). With this reporter, we found that loss of RIF1 did not obviously affect the frequency of No Indel EJ, although transient expression of RIF1 in the RIF1-KO caused a modest (1.4-fold) increase (Fig. 7A, B). In contrast, in cells treated with the M3814 DNA-PKcs inhibitor, loss of RIF1 caused a substantial (3.3-fold) decrease in No Indel EJ, which was recovered with transient expression of RIF1-WT (Fig. 7A, B). Namely, the fold effect of M3814 to inhibit No Indel EJ is magnified in RIF1-KO vs. parental and RIF-KO complemented with RIF1 (Supplementary Fig. 1). As a control, we found that M3814 treatment inhibits DNA-PKcs kinase activity similarly in RIF-KO and Parental cells (Supplementary Fig. 2). These findings indicate that RIF1 promotes No Indel EJ under conditions of DNA-PKcs inhibition, which is similar to the above findings with 53BP1.

We then used the MA-del assay to examine the effect of loss of RIF1 and DNA-PKcs inhibition on various EJ outcomes. We found that loss of RIF1, alone, and combined with DNA-PKcs inhibition had no significant effect on the categories of EJ junctions (i.e., No Indel, Insertion, or Deletion outcomes, Fig. 7C). Additionally, we also found that loss of RIF1 had no obvious effect on the pattern of Insertions (i.e., length and % consistency with Cas9 staggered DSBs, Fig. 7D). Thus, while loss of RIF1 combined with DNA-PKcs inhibition caused a reduction in No Indel EJ with the EJ7-GFP reporter, we did not observe this difference with the MA-del assay. However, this discrepancy is similar to the above findings with XLF and 53BP1, which again we suggest is due to the EJ7-GFP assay being more sensitive to changes in the efficiency of blunt DSB EJ.

## RIF1 loss causes a similar shift in deletion patterns as DNA-PKcs disruption

Beyond EJ categories per se, we next considered that RIF1 may affect the deletion pattern (i.e., size and/or microhomology use), as we found with 53BP1. From analyzing the deletions in the above experiment, we found that loss of RIF1 causes an increase in microhomology-associated −2, −7, and −75 deletions (Fig. 8A, B). Furthermore, M3814 treatment of RIF1-

KO cells caused a slight increase in −2 deletion size frequency, but not for the −7 or −75 deletions. (Fig. 8A). This pattern is very similar to that of loss of 53BP1, which itself is similar to disruption of DNA-PKcs (Fig. 3, Supplementary Fig. 4). Finally, from analysis of microhomology use from deletions, we found that loss of RIF1 causes an increase in microhomology that is similar to M3814 treatment, and combined RIF1 loss with M3814 treatment is similar to RIF1 loss alone (Supplementary Fig. 7). These findings indicate that RIF1 has a similar role to limit microhomology-associated deletions as 53BP1 and DNA-PKcs.

As a control for all MA-del assay experiments, we also examined the total frequency of the deletion rearrangement, using a quantitative PCR (qPCR) assay. Namely, we performed qPCR of the MA-del deletion rearrangement, normalized to a control PCR in the *MTAP* locus, each quantified using a fluorescent probe. As expected, we found that the qPCR signal for the MA-deletion rearrangement was dependent on transfection of the Cas9/sgRNAs that target the *MTAP* and *CDKN2B-AS1* loci (>80-fold increase, Supplementary Fig. 8). Notably, none of the M3814 treatments and genetic disruptions caused a significant decrease in deletion frequency (Supplementary Fig. 8). Although, a few treatments (e.g., M3814 treatment) and genetic disruptions caused an increase in deletion frequency, compared to the Parental control (Supplementary Fig. 8). Notably, this increase in deletion frequency via M3814 treatment is consistent with a report showing that low dose DNA-PKcs inhibition causes an increase in translocations[58]. In summary, the deletion frequency was readily detected and not obviously reduced with all the variants of genetic disruptions and M3814 treatments.

## Influence of 53BP1, RIF1, and DNA-PKcs on HDR and radiosensitivity

Finally, we examined other aspects of DSB repair: gene editing via HDR and radiosensitivity. Inhibition of DNA-PKcs has been shown to cause elevated HDR and radiosensitization, which have clinical applications[47,49,59,60]. Thus, we sought to define how 53BP1 and RIF1 affect the relative influence of DNA-PKcs disruption on these processes.

To examine HDR, we used the LMNA-HDR assay, which involves inducing a Cas9 DSB at the *LMNA* gene along with a donor template plasmid, which if used for HDR causes an *LMNA-mRuby2* fusion gene, encoding a fluorescent protein that can be measured by flow cytometry (Fig. 9A). Using this assay, we found that M3814 treatment alone caused a substantial increase in HDR (2.8-fold), whereas genetic loss of DNA-PKcs caused a mild increase (1.3-fold), similar to prior studies[47] (Fig. 9B). Loss of 53BP1 caused an increase in HDR in DNA-PKcs proficient cells (4.1-fold, i.e. 53BP1-KO vs. parental), in M3814-treated cells (2.2-fold, i.e., 53BP1-KO M3814 vs. parental M3814), and in cells with DNA-PKcs loss (1.6-fold, i.e., 53BP1-KO/PRKDC-KO vs. PRKDC-KO, Fig. 9B). Transient expression of 53BP1 caused a decrease in HDR in each of these scenarios. Thus, 53BP1 inhibits HDR in both DNA-PKcs proficient and deficient cells, although in the latter, the fold effect of 53BP1 loss on HDR is diminished (Fig. 9B). Notably, the HDR frequency for 53BP1-KO/PRKDC-KO, while higher than PRKDC-KO, was lower than 53BP1-KO. Thus, without 53BP1, loss of DNA-PKcs appears to cause a decrease in HDR. These findings underscore the difference between DNA-PKcs loss vs. kinase inhibition[61].

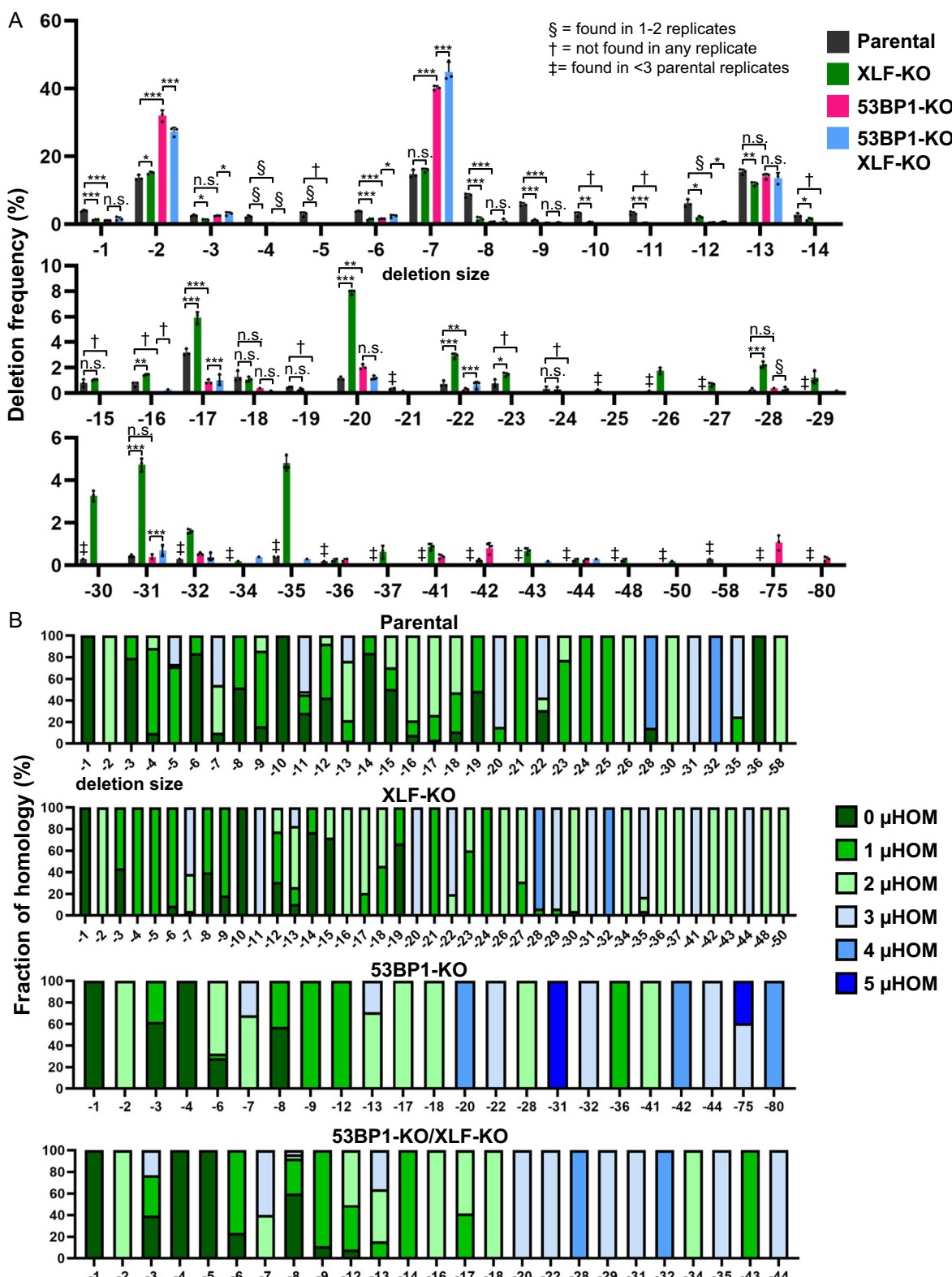

**Fig. 6 | Loss of 53BP1 and XLF causes distinct deletion patterns, and the effect of 53BP1 dominates in the double mutant. A** Combined loss of 53BP1 and XLF causes an increase in −2 and −7 deletions, similar to the loss of 53BP1 alone. Shown are deletion sizes for the samples shown in Fig. 5C. Frequencies represent the mean ± SD. $n = 3$ biologically independent transfections. Statistics with an unpaired *t*-test using Holm–Sidak correction. \*$P < 0.05$, \*\*$P < 0.01$, \*\*\*$P < 0.001$, \*\*\*\*$P < 0.0001$, n.s. = not significant. § = deletion size was only found in 1 or 2 replicates, † = deletion size was not found in any of the replicates, and ‡ = deletion size was found in <2 Parental replicates. **B** Microhomology use varies with distinct deletions. Shown is the fraction of microhomology used for each deletion size and experimental condition shown in (Fig. 5C). $n = 3$ biologically independent transfections. Source data are provided as a Source data file.

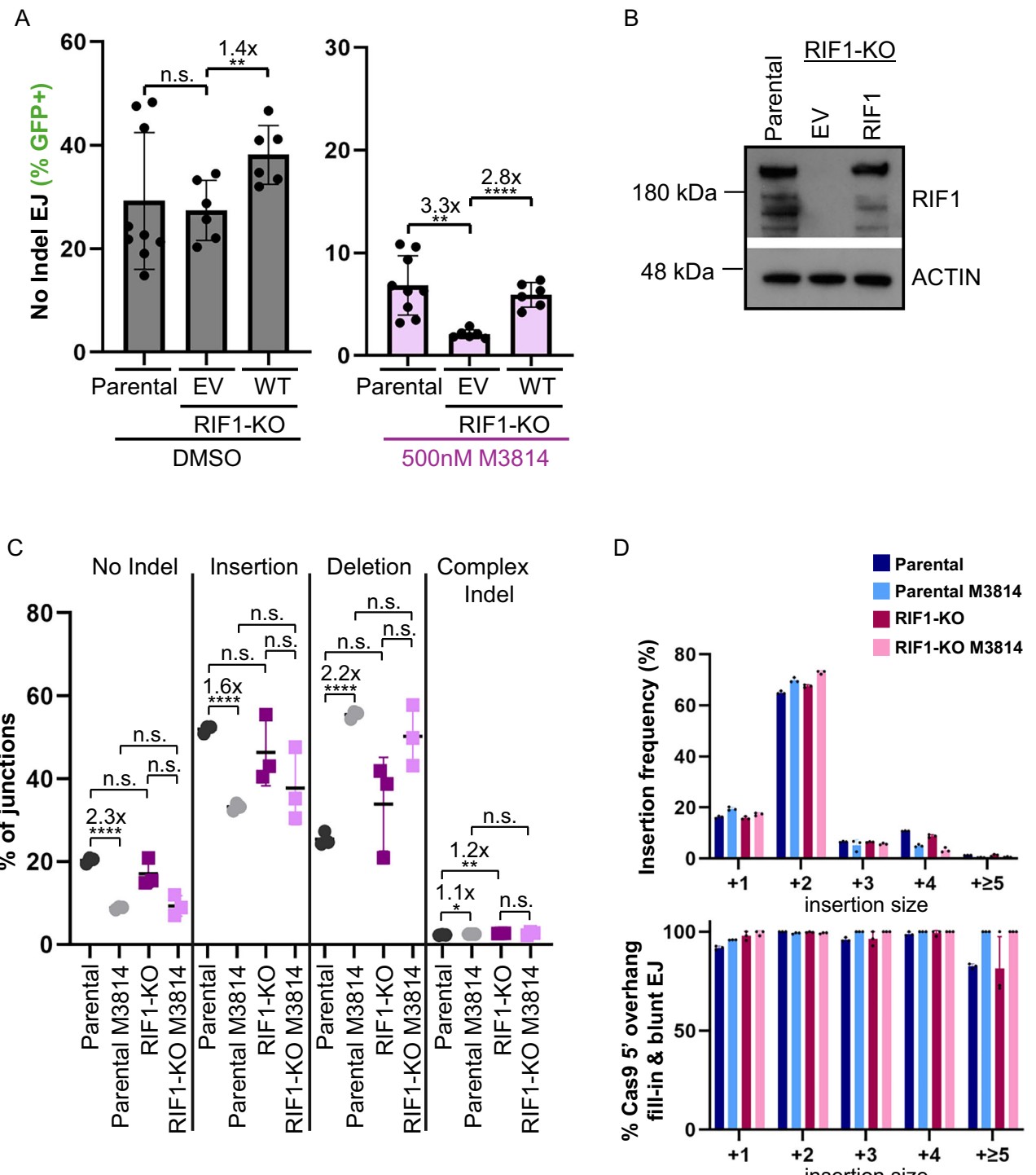

**Fig. 7 | Combined loss of RIF1 and DNA-PKcs inhibition causes a decrease in No Indel EJ. A** Loss of RIF1 alone is not important for No Indel EJ, but becomes important with the inhibition of DNA-PKcs. Frequencies represent the mean ± SD. $n = 6$ biologically independent transfections, except Parental $n = 9$. Statistics with an unpaired $t$-test using Holm−Sidak correction. **left to right $P = 0.0089$, $P = 0.0016$, ****$P < 0.0001$, n.s. = not significant. Parental values are those seen in Fig. 1B. **B** Immunoblot analysis of RIF1 for the cell lines and conditions shown in (**A**). **C** Loss of RIF1 alone has no obvious effect on EJ junction patterns. Shown are junction frequencies for No Indel EJ, Insertions, Deletions, and Complex Indels for

each condition. Frequencies represent the mean ± SD. $n = 3$ biologically independent transfections. Statistics with an unpaired $t$-test using Holm−Sidak correction. *$P = 0.006465$, **$P = 0.000807$, ****$P > 0.0001$, n.s. = not significant. Parental values are those seen in Fig. 2C. **D** Insertion events are likely associated with Cas9 5′ overhang fill-in and blunt DNA EJ in all conditions. Shown is the frequency of Insertion sizes and percent Insertions consistent with Cas9 5′ overhang fill-in and blunt EJ for the experimental conditions shown in (**C**). Frequencies represent the mean ± SD. $n = 3$ biologically independent transfections. Source data are provided as a Source data file.

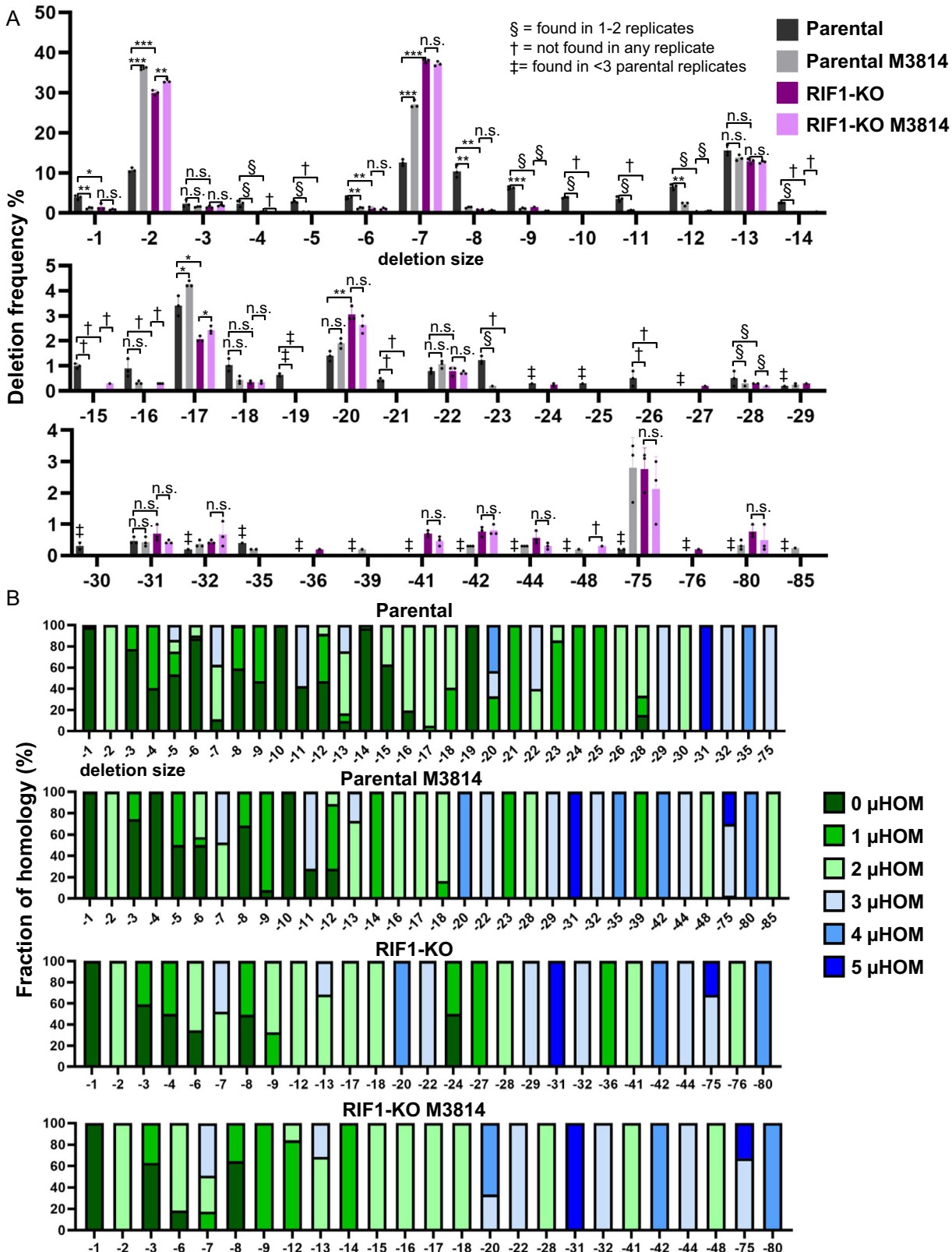

**Fig. 8 | RIF1 loss causes a similar shift in deletion patterns as DNA-PKcs disruption. A** Loss of RIF1 causes an increase in −2 and −7 deletions, similar to inhibition of DNA-PKcs. Shown are deletion sizes for the samples shown in (Fig. 7C). Frequencies represent the mean ± SD. $n = 3$ biologically independent transfections. Statistics with an unpaired $t$-test using Holm−Sidak correction. *$P < 0.05$, **$P < 0.01$, ***$P < 0.001$, ****$P < 0.0001$, n.s. = not significant. Parental values are those seen in (Fig. 3A). § = deletion size was only found in 1 or 2 replicates, † = deletion size was not found in any of the replicates, and ‡ = deletion size was found in <2 Parental replicates. **B** Microhomology use varies with distinct deletions. Shown is the fraction of homology used for each deletion size and experimental condition shown in (Fig. 7C). $N = 3$ biologically independent transfections. Parental values are those seen in (Fig. 3B). Source data are provided as a Source data file.

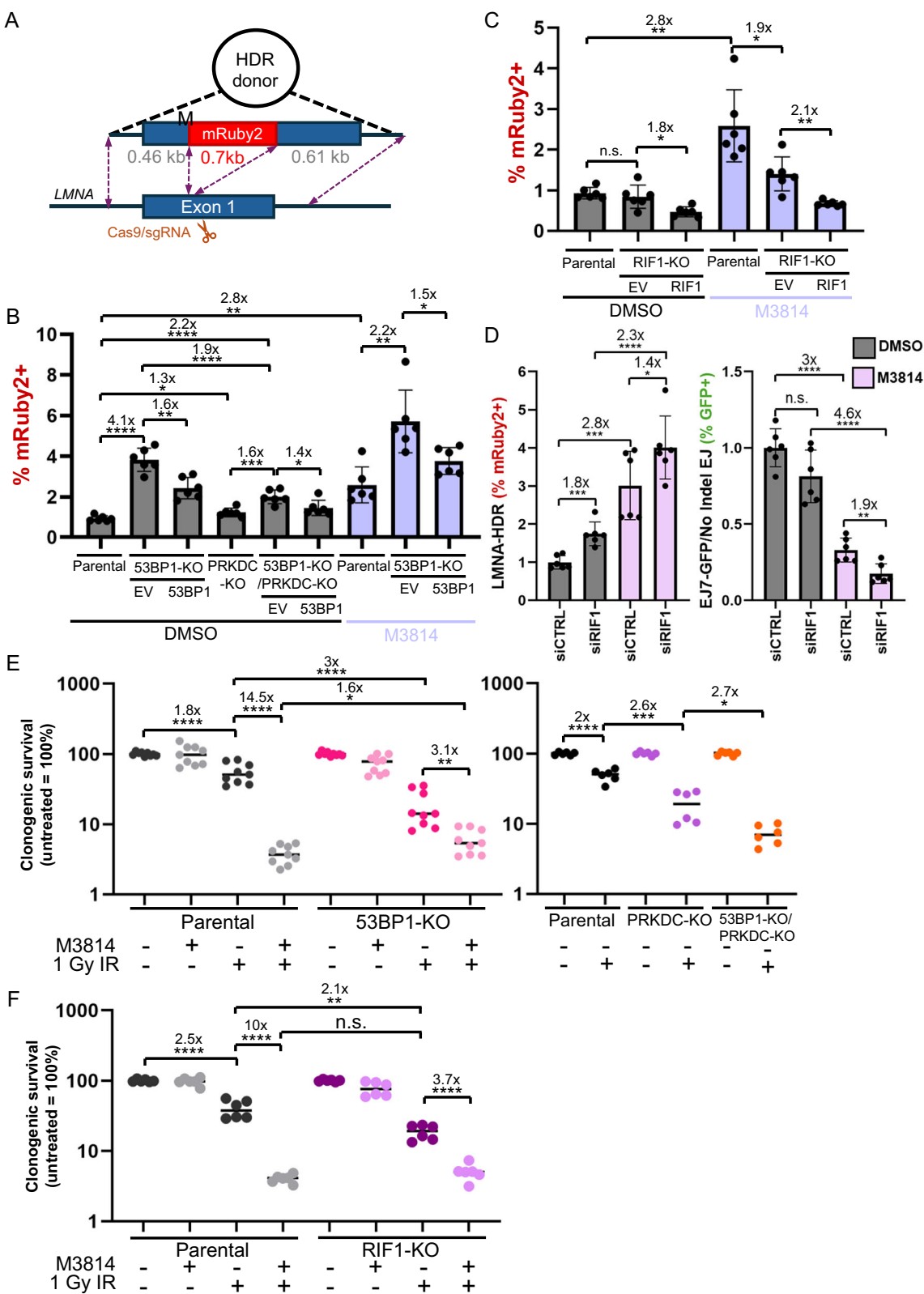

For RIF1, genetic loss alone had no substantial effect on HDR, yet transient expression of RIF1 in RIF1-KO caused a decrease in HDR both with and without M3814 (Fig. 9C). To address this discrepancy of finding no substantial effect on HDR in the RIF1-KO, yet expression of RIF1 in these cells caused a decrease in HDR, we tested RIF1 disruption using another method. Specifically, we examined depletion of RIF1 via siRNA (siRIF1 treatment). From these experiments, we found that

siRIF1 treatment caused an increase in HDR both with and without M3814, although the fold effect with M3814 was lower (1.8-fold vs 1.4-fold, Fig. 9D, Supplementary Fig. 5). This finding is similar to that of loss of 53BP1. For comparison with this HDR experiment, we also examined No Indel EJ using the EJ7-GFP reporter. We found that siRIF1 treatment failed to cause a significant decrease in No Indel EJ without M3814, yet caused a significant decrease with M3814 treatment (1.9-

**Fig. 9 | Influence of 53BP1, RIF1, and DNA-PKcs on HDR and radiosensitivity.**
**A** Schematic of the LMNA-HDR reporter that involves a Cas9/sgRNA expression vector that induces a single DSB in *LMNA* exon 1, as well as a plasmid donor, where HDR yields expression of mRuby2 from the *LMNA* locus. mRuby2 frequencies are normalized to transfection efficiency with parallel GFP transfections. **B** 53BP1 loss causes an increase in HDR in both DNA-PKcs proficient and deficient cells. Frequencies represent the mean ± SD. $n = 6$ biologically independent transfections. Statistics with an unpaired *t*-test using Holm–Sidak correction. *left to right $P = 0.0178$, $P = 0.0251$, $P = 0.017523$, **left to right $P = 0.0013$, $P = 0.0011$, $P = 0.001566$, ***$P = 0.0008$, ****$P < 0.0001$. **C** RIF1 genetic loss does not obviously increase HDR, but RIF1 expression inhibits HDR independent of DNA-PKcs kinase inhibition. Frequencies represent the mean ± SD. $n = 6$ biologically independent transfections. Statistics with an unpaired *t*-test using Holm–Sidak correction. *left to right $P = 0.0151$, $P = 0.0145$, **left to right $P = 0.0011$, $P = 0.0018$, n.s. = not significant. **D** Depletion of RIF1 via siRNA (siRIF1 vs. non-targeting siCTRL) causes an increase in HDR independent of DNA-PKcs kinase inhibition (LMNA-HDR assay),

and in contrast causes a decrease in No Indel EJ (EJ7-GFP) specifically in M3814-treated cells. Frequencies represent the mean ± SD. $n = 6$ biologically independent transfections. Statistics with an unpaired *t*-test using Holm–Sidak correction. *$P = 0.0399$, **$P = 0.0039$, ***left to right $P = 0.0004$, $P = 0.0005$, ****$P < 0.0001$, n.s. = not significant. **E** 53BP1 loss causes radiosensitivity that is less than DNA-PKcs kinase inhibition, but similar to DNA-PKcs loss. Cells were treated with M3814 or vehicle control (DMSO), and 0 Gy or 1 Gy IR dose, and plated to form colonies. Fraction clonogenic survival was determined relative to the untreated control (DMSO 0 Gy) for each cell line. Frequencies represent the mean ± SD. $n = 6$ independent wells of plated cells. Statistics with an unpaired *t*-test using Holm–Sidak correction. *left to right $P = 0.0257$, $P = 0.0124$, **$P = 0.004$, ***$P = 0.0003$, ****$P > 0.0001$. **F** Loss of RIF1 causes radiosensitivity that is less than DNA-PKcs kinase inhibition. Cells were treated and analyzed as in (**E**). Frequencies represent the mean ± SD. $n = 6$ independent wells of plated cells. **$P = 0.002064$, ****$P > 0.0001$, n.s. = not significant. Source data are provided as a Source data file.

fold, Fig. 9D). These effects with siRIF1 treatment are similar to those of RIF1 loss, and hence confirm that RIF1 is important for No Indel EJ when combined with DNA-PKcs inhibition. We suggest that the influence of RIF1 on HDR and No Indel EJ is similar to 53BP1.

Lastly, we examined clonogenic survival (colony formation) following ionizing radiation (IR). We selected one dose of IR (1 Gy), based on prior studies showing significant radiosensitization via M3814 in HEK293 cells at this dose[47]. Indeed, M3814 treatment of Parental cells caused a marked hypersensitivity to IR (14.5-fold, Fig. 9E). Loss of 53BP1 alone caused a 3-fold hypersensitivity to IR, compared to Parental cells, and treating 53BP1-KO cells with M3814 caused a further 3.1-fold hypersensitivity to IR (Fig. 9E). Thus, while M3814 treatment caused hypersensitivity to 53BP1-KO cells, the fold effect was less as compared to Parental cells. Indeed, 53BP1-KO cells treated with M3814 were only modestly radiosensitive compared to M3814-treated Parental cells (1.6-fold, Fig. 9E). We then tested combined genetic loss of 53BP1 and DNA-PKcs (PRKDC-KO), finding that PRKDC-KO cells show a 2.6-fold hypersensitivity to IR vs. Parental cells, and the double mutant (53BP1-KO/PRKDC-KO) was a further 2.7-fold hypersensitive compared to the PRKDC-KO single mutant (Fig. 9E). Loss of RIF1 alone caused a 2.1-fold hypersensitivity to IR, and M3814 treatment of RIF1-KO cells caused a further 3.7-fold hypersensitivity (Fig. 9F). In summary, loss of 53BP1 and RIF1 cause IR sensitivity, and while addition of M3814 treatment causes further radiosensitivity in these cells, the effects are not additive. In contrast, combining genetic loss of 53BP1 and DNA-PKcs causes approximately additive hypersensitivity to IR.

## Discussion

53BP1 is a key DNA damage response factor recruited to DSBs through interactions with chromatin marks that flank DSBs[28]. 53BP1 has been shown to affect DSB repair by inhibiting HDR, particularly in BRCA1-deficient cells, mediating class switch recombination during antibody maturation, as well as promoting fusion of deprotected telomeres[22-28]. However, the influence of this factor on affecting diverse EJ outcomes, and its functions in DSB repair in relation to NHEJ, have been unclear. We have focused on the genetic relationship between 53BP1 and DNA-PKcs, because both factors have been implicated in DSB end synapsis and regulation of DSB end processing[3-7,22-28,41,62]. Furthermore, defining the interplay with DNA-PKcs is significant, as DNA-PKcs kinase inhibitors are in clinical development[47,49,59,60]. We also examined RIF1, which is an effector protein of 53BP1[28-39].

From analysis of blunt DSB EJ, which is dependent on several NHEJ factors (e.g., XRCC4), we found that 53BP1 is dispensable but plays a backup role in cells disrupted for DNA-PKcs (both kinase inhibition and genetic loss, Fig. 10). This decrease in blunt DSB EJ is associated with a shift toward deletion mutations. Thus, 53BP1 appears to promote blunt DSB EJ and suppress EJ involving end processing, but these roles are masked by the function of DNA-PKcs. DNA-PKcs promotes DSB end

synapsis in a long-range complex via interactions with the DNA-PKcs dimer and interactions with Ku80 on the opposing DSB end[5-8]. Subsequently, this long-range complex can transition to a short-range complex without DNA-PKcs[5-8]. In this short-range complex, DSB ends are positioned for ligation[5-8]. We suggest that 53BP1 can also assist with end synapsis to stabilize the long-range complex and/or short-range complex but serves as a backup to synapsis mediated by DNA-PKcs. We also found that 53BP1 appears to serve as a key backup for end synapsis mediated by XLF. As a possible mechanism for mediating end synapsis, 53BP1 can oligomerize and form large condensates[42], as well as form microdomains that are important for chromosome condensation at DSBs[41]. Alternatively, 53BP1 could mediate direct interactions with the NHEJ complex to support end synapsis.

While 53BP1 loss did not affect the frequency of deletion mutations, it had a substantial effect on the type of deletions. Specifically, loss of 53BP1 caused an increase in deletions with microhomology, and conversely, a decrease in those without microhomology (Fig. 10). In this case, the effect of 53BP1 loss was similar to that of DNA-PKcs disruption (both genetic loss and kinase inhibition), and the combined disruption did not have an additive effect. Thus, 53BP1 and DNA-PKcs appear to function in the same pathway to affect the type of deletions. Our findings are consistent with a recent report that 53BP1 loss (in *Ku70−/−* mouse cells) causes an increase in microhomology usage during V-J recombination[63]. We speculate that 53BP1 and DNA-PKcs could be important to promote end processing events that are tightly linked to completion of repair by NHEJ. This model is supported by studies of the kinetics of NHEJ repair of IR-induced damage in G1, which appears to occur in a fast phase and a slow phase, the latter of which may involve DSB end processing[64]. Notably, while XLF loss also caused an increase in microhomology deletions, the deletion sizes were distinct from those affected by DNA-PKcs and 53BP1[53]. Furthermore, with the combined loss of 53BP1 and XLF, the deletion pattern caused by the loss of 53BP1 appears to dominate. The mechanisms that drive these distinct microhomology deletion patterns are unclear. We suggest that further defining the factors that influence the type of deletion mutation during EJ will provide insight into such mechanisms.

The effects of disrupting 53BP1 on EJ were similar of loss of RIF1, which is a direct effector protein that binds phosphorylated 53BP1[28-39]. For one, genetic loss of RIF1, as well as RNAi depletion of RIF1, caused a reduction in No Indel EJ when combined with DNA-PKcs kinase inhibition. We suggest that RIF1, like 53BP1, mediates end synapsis as a backup to DNA-PKcs (Fig. 10). Consistent with a role in end synapsis, RIF1 appears to stabilize condensed chromatin at DSBs and promote 53BP1 macrodomains that may be involved in such stabilization[41]. Loss of RIF1 also yielded a shift to microhomology deletions that is similar to loss of 53BP1 and disruption of DNA-PKcs. Thus, as with 53BP1, RIF1 appears to suppress microhomology EJ deletions, which is consistent with the notion that RIF1 is an effector protein of 53BP1[28-39]. As

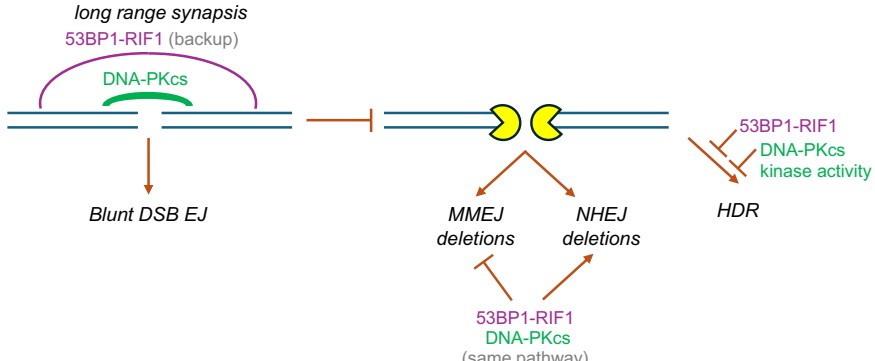

**Fig. 10 | Model of genetic relationship between 53BP1-RIF1 and DNA-PKcs on diverse DSB repair outcomes.**

mentioned above, defining additional factors that affect the type of deletion mutation will provide insight into the mechanism of 53BP1-RIF1-mediated suppression of microhomology EJ deletions.

We also examined the influence of 53BP1, RIF1, and DNA-PKcs on HDR. Loss of 53BP1 caused an increase in HDR in both DNA-PKcs proficient and deficient cells. Conversely, DNA-PKcs kinase inhibition also caused an increase in HDR in both 53BP1 proficient and deficient cells. However, the fold effects of 53BP1 loss and DNA-PKcs kinase inhibition were less than additive, which might indicate a partial overlap in function. For example, loss of 53BP1 and DNA-PKcs kinase inhibition might cause an increase in HDR via distinct mechanisms, yet the combination could cause a maximal possible increase in HDR. Findings with RIF1 depletion via siRNA were similar to 53BP1 loss. Interestingly, DNA-PKcs kinase inhibition caused a much greater increase in HDR vs. genetic loss, which reflects that kinase inhibition does not always have the same effect as loss. Indeed, DNA-PKcs kinase inhibition causes a more severe mouse phenotype vs. the genetic knockout, likely because DNA-PKcs kinase inhibition stabilizes its interaction with DNA ends[61]. Indeed, these findings with HDR are consistent with reports that DNA-PKcs bound to DNA ends is a signal for initiation of DSB end resection via the MRE11 complex and CtIP[62]. Specifically, the addition of DNA-PK (DNA-PKcs and KU) caused activation of nuclease activity of the MRE11 complex along with CtIP that was phosphorylated during purification[62]. Furthermore, this nuclease activity was enhanced with the inhibition of DNA-PKcs kinase activity that stabilizes its binding to DNA ends[62].

Finally, we examined the influence of disrupting 53BP1, RIF1, and DNA-PKcs on radiosensitivity. We found that DNA-PKcs kinase inhibition caused marked radiosensitivity, which was greater than the radiosensitivity caused by genetic loss of DNA-PKcs. Furthermore, the effects of genetic loss of DNA-PKcs, 53BP1, and RIF1 were similar to each other. Notably, we also found that DNA-PKcs kinase inhibition had the greatest overall effect on DSB repair outcomes: causing an increase in HDR, a decrease in blunt DSB EJ, and an increase in microhomology deletions. We speculate that the combination of these diverse effects on DSB repair outcomes contributes to the substantial radiosensitivity caused by DNA-PKcs kinase inhibition. Namely, these defects in EJ could cause toxic mutations and chromosomal rearrangements. Furthermore, elevated HDR could cause repair before sister chromatid synthesis, resulting in the use of ectopic donors that can also cause chromosomal rearrangements. In contrast, genetic loss of DNA-PKcs, 53BP1, and RIF1 each had relatively more modest effects on DSB repair outcomes vs. DNA-PKcs kinase inhibition. For example, loss of 53BP1 alone failed to cause a defect in blunt DSB EJ, but caused similar effects as DNA-PKcs kinase inhibition on causing an increase in microhomology deletions and HDR. Thus, we suggest that 53BP1-RIF1 and DNA-PKcs show distinct genetic interactions depending on specific DSB repair outcomes, which together are important for radioresistance.

We conclude with a discussion of various limitations of this study. For one, disruption of DNA-PKcs (loss or M3814 treatment) could affect not only NHEJ complexes, but also chromatin structure and/or regulation of cell cycle control, which could also play a role in DSB repair outcomes. Namely, DNA-PKcs has been shown to promote chromatin condensation, chromatin compartmentalization, and regulation of DNA replication in response to DNA damage[65–67]. Furthermore, since cyclin-dependent kinases (CDKs) regulate DSB end resection[68], effects of genetic knockouts or M3814 treatment on CDK regulation could also affect repair outcomes. Similarly, M3814 treatment could also have potential off-target effects on other kinases, although kinase inhibition studies, as well as structural studies, indicate that M3814 is highly specific for DNA-PKcs[48,49]. The genetic complementation assays involve transient expression, and the limitation of this approach is that complementation levels can vary between cells and often do not precisely match the endogenous levels.

The Cas9 approach to studying DSB repair has some inherent limitations. For one, genetic assays for DSB repair require a genetic change, also known as repair scars, and hence cannot detect precise repair of DSBs. To partially mitigate this issue, we examine the repair of two tandem blunt Cas9 DSBs without indels, but of course, this is not the same as examining the precise repair of a single DSB. Furthermore, such assays require that the tandem Cas9 DSBs occur simultaneously, which may not always be the case. Also, Cas9 DSBs can either be blunt or staggered, which, as we have shown, can complicate the analysis of EJ junctions. The relative probability of Cas9 to generate blunt DSBs vs. staggered DSBs, as well as the number of nucleotides of the 5′ overhang, appears to be affected substantially by sequence context[54,69]. Also, since blunt DSBs that are re-ligated restore the recognition site, and hence are prone to repeated cycles of DSB formation, it is difficult to determine the frequency of staggered vs. blunt DSBs for each given cleavage cycle. Nonetheless, for the MA-del assay, due to the Protospacer Adjacent Motif (PAM) orientation, staggered DSBs cause insertions from either DSB (Supplementary Fig. 3). For example, a 1 nt insertion could be caused by a 1 nt staggered DSB either in the *MTAP* locus or the *CDKN2B-AS1* locus. We suggest that listing such limitations supports further research in this area using these approaches.

## Methods
### Cell lines and plasmids
For sgRNA and Cas9 expression, the px330 plasmid was used (deposited by Dr. Feng Zhang, Addgene 42230)[70]. All sequences for the sgRNAs used in this study, including the sgRNAs for the EJ7-GFP assay (7a and 7b), and the MA-del assay (MTAP and CDKN2B-AS1), are found in Supplementary Table 1. The pCAGGS-53BP1 expression vector was described previously[50], and the pCAGGS-RIF1 expression vector used the RIF1 coding sequence from pDEST pcDNA5-FRT/TO-eGFP-RIF1 (deposited by Dr. Daniel Durocher, Addgene 52506)[30], and the empty vector control is pCAGGS-BSKX[71]. Plasmids for the LMNA-HDR assay

(LMNA Cas9/sgRNA and LMNA-mRuby2-Donor plasmids) were provided by Dr. Jean-Yves Masson and were previously described[72].

The HEK293 EJ7-GFP cell line, the XLF-KO, and PRKDC-KO cells were previously described[47,57]. 53BP1-KO, RIF1-KO, 53BP1-KO/XLF-KO, and 53BP1-KO/PRKDC-KO cell lines were generated using Cas9/sgRNAs targeting 53BP1 and RIF1, each using one sgRNA for RIF1, and two sgRNAs for 53BP1. To generate the knockout cell lines, cells were co-transfected with the Cas9/sgRNA and pgk-puro plasmid, transfected cells were enriched using puromycin treatment, and were then plated at low density to isolate and screen individual clones for genetic disruption via immunoblotting.

## DSB reporter assays

To test the EJ7-GFP reporter assay, HEK293 cells were seeded in a 24-well plate coated with poly-lysine at $0.5 \times 10^5$ cells/well. The next day, cells were transfected with 200 ng of sgRNA/Cas9 plasmids (7a and 7b), and 50 ng of 53BP1, RIF1, or EV control plasmid. To test transfection efficiency, parallel transfections were performed with 200 ng of GFP-expressing plasmid (pCAGGS-NZE-GFP) and 200 ng of EV with respective amounts (50 ng) of 53BP1, RIF1, and EV. For each well, all transfections used 1.8 µL of Lipofectamine 2000 (Thermo Fisher) and 0.5 mL of antibiotic-free media. To test the LMNA-HDR assay, LMNA Cas9/sgRNA and LMNA-mRuby2-Donor plasmids were used in place of the two sgRNA/Cas9 plasmids (200 ng each). Cells were incubated for 4 h with the transfection agents, washed, and treated with media containing 500 nM M3814 (i.e., Nedisertib, Selleckchem #S8586) or vehicle (Dimethyl Sulfoxide, DMSO). For siRNA experiments, cells were seeded on 24-well plates with transfection complexes with 5 pmol of siRNA and 1.8 µL of Lipofectamine RNAiMAX (Thermofisher), and the plasmid transfection also included 5 pmol of siRNA (siCTRL Dharmacon D-001810-01, siRIF1 Dharmacon siGENOME SMARTpool mixture of D-027983-01, -02, -03, and -04). All wells had the same total amount of DMSO in each experiment. 3 days after the transfection, cells were analyzed using flow cytometry (ACEA Quanteon, Agilent NovoExpress Version 1.5.0), as described (Supplementary Fig. 9)[47].

For the MA-del assay, cells were seeded in a 6-well plate coated with poly-lysine. Cells were transfected with 800 ng each of MTAP and CDK2NB1-AS1 Cas9/sgRNA plasmids, as well as 400 ng of pgk-puro plasmid. Each well used 7.2 µL of Lipofectamine 2000 and 2 mL of antibiotic-free media. As in the EJ7 assay, cells were incubated for 4 h with transfection agents, washed, and treated with media containing M3814 or DMSO. The day after transfection, cells were plated into M3814- or DMSO-containing complete media with 5 µg/mL puromycin for 2 days to enrich for transfected cells via puromycin selection and then expanded into media without puromycin for 3 days. Genomic DNA isolation was conducted as described[71]. PCR amplification (Platinum HiFi Supermix, Thermo Fisher) of the MTAP-CDK2NB1 rearrangement used the MAfusion1UP and MAfusion1DN primers.

The amplicons were subjected to deep sequencing using the Amplicon-EZ service (Azenta) and their SNP/INDEL detection pipeline, which uses the Burrows-Wheeler Aligner (BWA) to align the reads to the predicted No Indel EJ junction sequence based on the sgRNA/Cas9 cut sites. Since this method uses targeted amplicon sequencing (primers that anneal to fixed positions in genomic DNA, such that amplicons begin/end with the same sequence), duplicate filtering is not performed. The indel categories were identified subsequent to this alignment step: WT and base changes (No Indel EJ product), deletions (continuous or discontinuous, and with or without base changes), Insertions (inserted nucleotides without deletions, and with or without base changes), and insertions and deletions (both Insertion and deletion, and with or without base changes). From this analysis, EJ outcomes were categorized as No Indel EJ, Insertions, Deletions, or Complex Indels. Then, the total reads in each category were used to assess their frequency (Supplementary Table 2, Supplementary Data

File 1, Supplementary Data File 2). Insertion and deletion sequences were analyzed individually by manual alignment, for all read sequences representing at least 0.1% of the combination of Insertions and complex indels, and at least 0.1% of the deletion reads, respectively. Some deletion events showed nucleotides consistent with staggered Cas9 cleavage, which were used to assign breakpoints for deletion size and microhomology. For each condition, amplicon sequencing was performed on 3 independent transfections.

To quantify the deletion rearrangement by qPCR, two sets of reactions were performed, both using the probe primer (MAfusionHYB), and with MAfusion1UP and MAfusion1DN for the deletion rearrangement, and MAfusion1UP and MTAPctrlDN1 for the control reaction. The qPCR was performed with iTaq Universal Probes Supermix and the CRX Connect Real-Time PCR Detection System (BioRad). The relative levels of the deletion rearrangement were determined using the cycle threshold (Ct) value for the deletion rearrangement reaction and then subtracted from the average Ct value for the control reaction (ΔCt). This value was then normalized to the ΔCt value from Parental cells analyzed in parallel, to calculate the 2−ΔΔCt value.

## Immunoblotting

To conduct immunoblot analysis, cells were seeded in a poly-lysine-coated 6-well plate and were transfected as previously described in the reporter assays, with the exception that the EV plasmid (pCAGGS-BSKX) was used instead of the sgRNA/Cas9 plasmid(s). Transfected cells were scraped from each well, lysed with ELB (250 mM NaCl, 5 mM EDTA, 50 mM Hepes, 0.1% (v/v) Ipegal, and Roche protease inhibitor), and then were sonicated (Qsonica, Q800R). To test for DNA-PKcs-S2056p, cells were treated with M3814 or DMSO for 3 h, treated with 10 Gy IR (MultiRad 160) treatment, and allowed to recover for 1 h. Protein was extracted using the ELB solution described above with the addition of PhosSTOP (Roche) and 50 µM sodium fluoride. Blots were probed with antibodies for 53BP1 (Abcam ab36823, 1:1000), RIF1 (Cell Signaling Technologies 95558s, 1:1000), DNA-PKcs (Invitrogen MA5-13238, 1:1000), DNA-PKcs-S2056p (Abcam ab124918, 1:1000), XLF (Bethyl A300-730, 1:1000), MLH1 (Abcam ab92312, 1:1000), and ACTIN (Sigma A2066, 1:3000). HRP signals were developed using ECL reagent (Amersham Biosciences).

## Clonogenic survival

To test clonogenic survival, 6-well plates were coated with poly-lysine, and cells were plated at various cell densities with 2.5 mL complete media containing M3814 or DMSO. The following day, cells were treated with 1 Gy IR (MultiRad 160) or left untreated. All cells (IR-treated and untreated) were treated with 3 mL fresh media containing M3814 or DMSO, and colonies were allowed to form for 7–10 days. Colonies were fixed in cold methanol before staining with 0.5% crystal violet (Sigma) in 25% methanol. Colonies were counted blindly, as in the sample identity was hidden from the experimenter conducting the counting, under a 4× objective. Survival was determined for each well relative to the mean value of DMSO/untreated wells for the respective cell line that was seeded in parallel.

## Reporting summary

Further information on research design is available in the Nature Portfolio Reporting Summary linked to this article.

## Data availability

The raw sequencing data with the fastq files for the MA-del assay have been deposited in the Sequence Read Archive at PRJNA1301407 and PRJNA1271093. The other datasets in the study are included in the main manuscript and it Supplementary files. Source data are provided with this study. Source data are provided with this paper.

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

## Acknowledgements

This study was funded in part by the National Cancer Institute of the National Institutes of Health: R01CA256989 (J.M.S.); P30CA33572 (City of Hope Core Facilities); F99CA284248 (M.C.A.).

## Author contributions

K.M., M.C.-A., F.W.L., and J.M.S. designed research; K.M., M.C.-A., and F.W.L. performed research; K.M. and J.M.S. analyzed data and wrote the paper with input from all authors.

## Competing interests

The authors declare no competing interests.
