## [Transparent Peer Review file · Nature Communications]

53BP1-RIF1 and DNA-PKcs show distinct genetic interactions with diverse chromosomal break repair outcomes

Corresponding Author: Professor Jeremy Stark

Version 0:

Reviewer comments:

Reviewer #1

(Remarks to the Author)

This is an interesting study using 293 cells with and without genetic KOs (and DNA-PKcs inhibitor) to study the effects on DNA end joining, with particular focus on NHEJ. This is an important topic with much yet to be learned. The Stark lab's approach is a valuable contribution. The authors' summaries of their data throughout the paper are useful in considering the structural biology and biochemistry within the NHEJ field. I have suggested numerous adjustments, which I fully expect the authors to take very seriously before I will fully endorse the paper.

Abstract: The use of the abbreviation EJ is confusing because readers may think they are referring to TMEJ or some other EJ processes when in fact, they are referring to alterations in NHEJ outcomes.

Intro: In the statement "In the absence of C-NHEJ factors, EJ remains relatively robust (citations 11-15)" the authors ignore the fact that KO of ligase 4 has a major impact on end joining. One paper cited (Corneo) may be complicated by persistent binding of RAG longer into the G1-S transition than normal and by the complexities of the X4 expression in the XR-1 cell line. Diploid yeast may be less vulnerable but haploid yeast with a L4 KO are very compromised. Drosophila has more TMEJ due to lack of some key NHEJ proteins.

RESULTS: The authors' results are valuable, despite some small effects. It would only enhance the paper if the authors would acknowledge the following major limitations, which should be included in a paragraph or two at the end of the Discussion about LIMITATIONS:

First, the M3814 inhibitor might be affecting other kinases that influence the cell cycle. This may alter the local chromatin at the assay gene sites in the two substrates used. Since 53BP1 and Rif1 may affect chromatin, this may influence the complex results observed. The fact that M3814 had a larger effect than any of the KOs may support this thought, which is not really diminish their study. Rather it opens new avenues for the future.

Second, could the subtle impact of the various KOs partly be due to various degrees of cell cycle effect?

Third, it was nice to see the MA-assay approach because, unlike the EJ7 assay, the MA-assay does not constrain the outcomes observed. But I did not see a clear description of the number of reads and how duplicates were handled. Since the authors emphasize the specific positions of micro homology (MH), many duplicates may be present. If so, the authors can simply acknowledge this, but at least they should mention it.

Fourth, what fraction of the reads or events are due to blunt cleavage by Cas9 and what fraction are due to staggered cleavage. In the Methods section and Results, this was not clear.

Fifth, the authors should point out that the two Cas9 breaks are not simultaneous. Therefore, this assay may not fully represent single site DSB events in natural circumstances, especially since the nucleosome proximity to the breaks at each of the two sites may be such that one break may occur hours before the other break.

Reviewer #2

(Remarks to the Author)

The proposed manuscript by Makins et al. investigates the genetic interplay between 53BP1 and canonical non-homologous end joining (C-NHEJ), with a particular focus on the DNA-PKcs kinase. The authors examine end-joining (EJ) repair of Cas9-induced blunt double-strand breaks (DSBs), which are primarily repaired via C-NHEJ. They report that loss of 53BP1 reduces blunt DSB repair only in the context of DNA-PKcs deficiency. Furthermore, the disruption of 53BP1/RIF1 and DNA-PKcs—individually or in combination—results in a similar increase in microhomology-mediated deletions. Based on these findings, the authors propose that 53BP1/RIF1 acts as a backup for DNA-PKcs during blunt DSB repair but functions in the same pathway to suppress microhomology-mediated deletions. In contrast, 53BP1 and DNA-PKcs appear to function independently in suppressing homology-directed repair (HDR). Finally, the authors show that DNA-PKcs kinase inhibition results in pronounced radiosensitivity, which is not further enhanced by the loss of 53BP1 or RIF1. Overall, the authors conclude that 53BP1/RIF1 and DNA-PKcs exhibit distinct genetic interactions influencing diverse DSB repair outcomes.

Given that 53BP1/RIF1 and DNA-PKcs are key regulators of DSB repair pathway choice, this study is of general interest to the DNA repair field and may have implications for chemotherapy response. However, concerns related to experimental design and data interpretation substantially weaken the impact of the findings.

Major Concerns:

Overexpression and Data Quality: Several critical experiments rely on overexpression systems or suffer from low data quality. This undermines confidence in the conclusions drawn from these experiments.

Reliance on DNA Repair Scars: The entire study is based on analysis of repair scars, without accounting for error-free repair outcomes that leave no detectable scars. This limits the interpretability of the conclusions regarding pathway utilization.

Oversimplification of DNA-PKcs Function: DNA-PKcs is not solely a C-NHEJ factor; it also promotes end resection through phosphorylation of CtIP, thereby facilitating HDR, SSA, and possibly MMEJ and shieldin-mediated NHEJ (Deshpande et al., *Science Advances*, 2020). This multifunctionality may explain some of the results shown in Figure 9 but is not adequately considered in the interpretation.

Incomplete Representation of 53BP1 Pathway: 53BP1 has multiple downstream effectors—DYNLL1, RIF1, shieldin, CST, and ASTE1—each potentially leading to distinct repair outcomes. The manuscript emphasizes RIF1 without providing side-by-side comparisons with 53BP1 or other effectors, yet presents RIF1 as uniquely aligned with 53BP1 in determining repair outcome. This oversimplifies the functional landscape of 53BP1 signaling.

Flawed Experimental Design (Figure 1A): The reporter design in Figure 1A fails to account for in-frame deletions, which could occur with up to 33% probability and still yield a fluorescent signal. The authors need to demonstrate that even single amino acid deletions at the cut site eliminate fluorescence, otherwise the assay may overestimate EJ efficiency.

High Variability in Key Data (Figures 1B and 7A): The DMSO control group in Figure 1B shows high variability, which compromises statistical conclusions. Notably, overexpression of 53BP1 increases scarless EJ regardless of DNA-PKcs status, contradicting the proposed model. Similarly, DNA-PKcs inhibition consistently decreases scarless EJ, regardless of 53BP1 expression, which again contradicts the author's interpretation.

Inconsistent Interpretation of DNA-PKcs and 53BP1 Interaction (Figure 2B): Inhibition of DNA-PKcs reduces insertion and scarless EJ while increasing deletions. Loss of 53BP1 exacerbates this shift toward deletions under DNA-PKcs inhibition. This pattern suggests additive effects, contradicting the conclusion that 53BP1 and DNA-PKcs act in the same pathway.

Confounded Overexpression Data (Figure 9B): Overexpression results are difficult to interpret due to variable transfection efficiency and increased protein levels. The data nonetheless suggest that DNA-PKcs promotes HR in 53BP1-KO cells—53BP1-KO increases HR by 4.1-fold, while combined 53BP1 and DNA-PKcs KO only results in a 1.6-fold increase—again challenging the authors' conclusion that these proteins act antagonistically in HDR suppression.

Most figures are well labeled and clearly presented, with the exception of Figures 1B and 7A. However, the underlying experimental design flaws and overly reductive data interpretation limit the study's rigor.

Reviewer #3

(Remarks to the Author)

Reviewer #4

(Remarks to the Author)

The authors of Makins et al investigated the roles of 53BP1/RIF1 and DNA-PKcs kinase activity on C-NHEJ using GFP reporter (No indel EJ) and locus-specific large scale deletion (MA-del). They found that although 53BP1 or RIF1 loss alone do not impact end joining, they have combined impact with DNA-PKcs kinase inhibition or deficiency or with XLF. An increase in microhomology deletions is also associated with repair perturbation. However, the cooperative effect does not extend into suppressing homology-directed repair and radiosensitivity as combined perturbations are further affected beyond single target perturbations. The work could be significantly improved by addressing the below comments.

Major Comments

Although the use of HEK293 cells to generate knockout cell lines is reasonable, there is some concern over the RIF1 deficiency phenotype, which was generated using only one gRNA instead of 2 gRNAs used for generating 53BP1 deficiency and is a standard for generating permanent deficiencies as a single gRNA could lead to codon in-frame reversion mutations and confound repair pattern significance. This is a well documented phenomenon for gene therapies prior to using Cas9 and is also why a second clone is often used to discern phenotypic variation, which this reviewer notes is lacking for any deficiency in this study. The speculation about the disparity between RIF1 and 53BP1 with regard to assay differences seems unnecessary, counterproductive to supporting the MA-del assay for future use, and potentially misleading for additional downstream experiments (e.g. qPCR). To resolve this central issue, further clones should be derived for RIF1 deficiency using a 2 gRNA setup and tested to confirm whether RIF1 deficiency has a true distinction with 53BP1 at the level of DSB repair.

Much of the results section go into the finer details of which sized deletion is enhanced, decreased or modified by junction structure, however, if it is not clear which of the deletion combinations for a given deletion size is used and how that may change with repair factor perturbation. Given the prominence of the data in main figures, more elaboration on drastically affected deletions should be provided with regard to how many deletion combinations represent the affecting deletion and whether this preference changes with repair perturbation (e.g., any further distinction of diversity within a specific deletion size for DNA-PKcs, 53BP1 and XLF). Perhaps a novel pattern independent of DSB position and sequence context may stand out and could be broadly applied for one, some or all of the tested repair factor perturbations.

The authors seemed to have missed a recent publication (PMID: 38412274) describing low dose DNA-PKcs kinase inhibition promoting translocations, which are consistently reproduced here using the qPCR assay in the manuscript. It would be important for the authors to contrast their findings with that recent publication to further substantiate the observations.

Minor Comments

Inclusion of X axis labels for like figure panels (e.g. Figure 3) would be useful to include.

Data is all presented as percentages and not clear how many junctions are available for analysis. The authors should list the numbers of sequence reads and junction breakdown for the major subset displayed in the figures.

Information for a sequence read repository of the raw sequencer data and access to processed data from the repository should be indicated.

Version 1:

Reviewer comments:

Reviewer #1

(Remarks to the Author)

Overall, I am supportive of the authors' changes except for a very minor aspect of the Introduction. I support the authors thinking about the roles of 53BP1, RIF1, Shieldin, and CST-Pola because I agree with the authors that the roles of these proteins need clarification in how they participate in NHEJ versus aEJ, SSA, or HR.

Prior to clarification of the distinction of NHEJ from aEJ (primarily due to TMEJ), the use of the designation C-NHEJ was understandable. But now that the distinction of NHEJ from aEJ is clearer, the prefix C- in front of NHEJ is unnecessary. I would suggest dropping the C- in front of NHEJ, as other recent reviews in this field have done.

I realize that the authors are grouping NHEJ and aEJ into one large group in the Introduction. While I am opposed to this, I can understand that the readout of the assays in this paper fit with that thinking. However, as the authors have now noted in the revision, blunt end joining in NHEJ is very infrequent due to local addition by pol mu and pol lambda and due to local limited resection by Artemis:DNA-PKcs. So blunt EJ is a very narrow view of NHEJ, which the authors acknowledge and which the authors call C-NHEJ.

If I could trouble the authors, I would suggest that they revise the couple of early paragraphs in the Intro to point out the uncertainties about 53BP1/RIF and point out that this paper was carried out to provide some insight. This would be preferable compared to the authors' current Intro which seems to want to reorganize the entire DSBR classification simply because of the uncertainties about what 53BP1 is doing.

I commend the authors' clarifications in their Responses overall, even in cases where I do not entirely agree. The authors should recognize, as said above, that most NHEJ (what they call C-NHEJ) is not precise and results in at least a few nucleotides of loss or addition. The study of precise events may account for part of their perspective on NHEJ versus other pathways.

Reviewer #2

(Remarks to the Author)

They have addressed all our concerns.

Reviewer #3

(Remarks to the Author)

Reviewer #4

(Remarks to the Author)

The authors have made substantial improvements to the manuscript that strengthen their conclusions. This reviewer believes the authors have satisfactorily addressed all of the reviewers' comments.

Point-by-point responses to the reviewers' comments for the manuscript "53BP1-RIF1 and DNA-PKcs show distinct genetic interactions with diverse chromosomal break repair outcomes." NCOMMS-25-36080 at *Nature Communications*.

We thank the reviewers for their positive comments and suggestions to improve the manuscript. We have responded to each concern with new experiments, analysis, and/or edits to the text. Each reviewer comment is shown in *italics/green*, and is followed by our response.

REVIEWER COMMENTS

Reviewer #1 (Remarks to the Author): This is an interesting study using 293 cells with and without genetic KOs (and DNA-PKcs inhibitor) to study the effects on DNA end joining, with particular focus on NHEJ. This is an important topic with much yet to be learned. The Stark lab's approach is a valuable contribution. The authors' summaries of their data throughout the paper are useful in considering the structural biology and biochemistry within the NHEJ field. I have suggested numerous adjustments, which I fully expect the authors to take very seriously before I will fully endorse the paper.

Abstract: The use of the abbreviation EJ is confusing because readers may think they are referring to TMEJ or some other EJ processes when in fact, they are referring to alterations in NHEJ outcomes.

Concern 1-1. There is concern with using the abbreviation EJ for end joining in the Abstract. We typically use "EJ" to encompass both C-NHEJ and ALT-EJ/TMEJ, but upon review of the Abstract, we agree with the reviewer that use of "EJ" is confusing in this setting. In response, we have replaced "EJ" with "NHEJ" in the Abstract, as requested. Furthermore, when we use "EJ" in the Introduction, we clarify that this is meant to encompass all EJ events, including C-NHEJ and ALT-EJ/TMEJ, as follows "There are several sub-pathways of NHEJ repair that differ by the factors involved, the possible substrates for ligation, and genetic outcomes. To avoid confusion between total NHEJ and canonical non-homologous end joining (C-NHEJ), we refer to all DSB end joining simply as EJ."

Intro: In the statement "In the absence of C-NHEJ factors, EJ remains relatively robust (citations 11-15)" the authors ignore the fact that KO of ligase 4 has a major impact on end joining. One paper cited (Corneo) may be complicated by persistent binding of RAG longer into the G1-S transition than normal and by the complexities of the X4 expression in the XR-1 cell line. Diploid yeast may be less vulnerable but haploid yeast with a L4 KO are very compromised. Drosophila has more TMEJ due to lack of some key NHEJ proteins.

Concern 1-2. There is a concern that the phrase "In the absence of C-NHEJ factors, EJ remains relatively robust," is not an accurate reflection of the literature. We agree. In response, we have made the following edits to the Introduction: "In the absence of C-NHEJ factors, there are backup EJ pathways that function to varying degrees based on the specific context (e.g., DSB end structure, organism, and cell type). Such backup EJ pathways appear to cause repair outcomes with greater microhomology at the junctions, indicating that microhomology annealing is involved in bridging the DSBs."

RESULTS: The authors' results are valuable, despite some small effects. It would only enhance the paper if the authors would acknowledge the following major limitations, which should be included in a paragraph or two at the end of the Discussion about LIMITATIONS:

First, the M3814 inhibitor might be affecting other kinases that influence the cell cycle. This may alter the local chromatin at the assay gene sites in the two substrates used. Since 53BP1 and Rif1 may affect chromatin, this may influence the complex results observed. The fact that M3814 had a larger effect than any of the KOs may support this thought, which is not really diminish their study. Rather it opens new avenues for the future.

Second, could the subtle impact of the various KOs partly be due to various degrees of cell cycle effect?

Concern 1-3. There is a recommendation to address a set of limitations of the study at the end of the Discussion. The concerns relate to possible off-target effects of M3814, and indirect effects of chromatin and cell cycle on DSB repair. There are also additional limitations listed below by this reviewer and others. In response, we have added two such paragraphs at the end of the Discussion, which begins “We conclude with a discussion of various limitations of this study.”

Third, it was nice to see the MA-assay approach because, unlike the EJ7 assay, the MA-assay does not constrain the outcomes observed. But I did not see a clear description of the number of reads and how duplicates were handled. Since the authors emphasize the specific positions of micro homology (MH), many duplicates may be present. If so, the authors can simply acknowledge this, but at least they should mention it.

Concern 1-4. There are two recommendations to improve the presentation of the deep-sequencing data from the MA-del assay.

1. Request to describe how duplicates are handled. Duplicate filtering is used for analysis of random ligation of adapters to sheared DNA, such that identical amplicons that start/end with the same sequence are likely caused by duplicates. However, filtering of duplicates is not feasible or appropriate for targeted amplicon sequencing. In response, we have added this sentence to the Methods “Since this method uses targeted amplicon sequencing (primers that anneal to fixed positions in genomic DNA, such that amplicons begin/end with the same sequence), duplicate filtering is not performed.”
2. Request to provide the raw numbers of reads. In response, we have added Supplementary Table 3 showing read counts for the categories of outcomes, and also have included un-analyzed raw excel files from the alignment as Supplementary materials (Supplementary Data File 1, Supplementary Data File 2).

Fourth, what fraction of the reads or events are due to blunt cleavage by Cas9 and what fraction are due to staggered cleavage. In the Methods section and Results, this was not clear.

Concern 1-5. There is a request to describe the relative probability of Cas9 generating blunt DSBs vs. staggered DSBs. We have responded by summarizing a detailed study

on this issue, as well as adding new Supplementary Figure 3 that illustrates the various overhang combinations in the MA-del assay. Specifically, we have added these sentences to the end of the Discussion:

- “The relative probability of Cas9 to generate blunt DSBs vs. staggered DSBs, as well as the number of nucleotides of the 5’ overhang, appears to be affected substantially by sequence context (references PMID 30033371, and PMID 37828409). Also, since blunt DSBs that are re-ligated restore the recognition site, and hence are prone to repeated cycles of DSB formation, it is difficult to determine the frequency of staggered vs. blunt DSBs for each given cleavage cycle. Nonetheless, for the MA-del assay, due to the Protospacer Adjacent Motif (PAM) orientation, staggered DSBs cause insertions from either DSB (Supplementary Figure 3). For example, a 1 nt insertion could be caused by a 1 nt staggered DSB either in the *MTAP* locus or the *CDKN2B-AS1* locus.”

Fifth, the authors should point out that the two Cas9 breaks are not simultaneous. Therefore, this assay may not fully represent single site DSB events in natural circumstances, especially since the nucleosome proximity to the breaks at each of the two sites may be such that one break may occur hours before the other break.

Concern 1-6. There is a request to note that for repair assays using 2 Cas9 DSBs, the two DSBs are not necessarily simultaneous. In response, we have added this limitation to the concluding Discussion paragraphs mentioned in the response to Concern 1-3.

Reviewer #2 (Remarks to the Author):

The proposed manuscript by Makins et al. investigates the genetic interplay between 53BP1 and canonical non-homologous end joining (C-NHEJ), with a particular focus on the DNA-PKcs kinase. The authors examine end-joining (EJ) repair of Cas9-induced blunt double-strand breaks (DSBs), which are primarily repaired via C-NHEJ. They report that loss of 53BP1 reduces blunt DSB repair only in the context of DNA-PKcs deficiency. Furthermore, the disruption of 53BP1/RIF1 and DNA-PKcs—individually or in combination—results in a similar increase in microhomology-mediated deletions. Based on these findings, the authors propose that 53BP1/RIF1 acts as a backup for DNA-PKcs during blunt DSB repair but functions in the same pathway to suppress microhomology-mediated deletions. In contrast, 53BP1 and DNA-PKcs appear to function independently in suppressing homology-directed repair (HDR). Finally, the authors show that DNA-PKcs kinase inhibition results in pronounced radiosensitivity, which is not further enhanced by the loss of 53BP1 or RIF1. Overall, the authors conclude that 53BP1/RIF1 and DNA-PKcs exhibit distinct genetic interactions influencing diverse DSB repair outcomes.

Given that 53BP1/RIF1 and DNA-PKcs are key regulators of DSB repair pathway choice, this study is of general interest to the DNA repair field and may have implications for chemotherapy response. However, concerns related to experimental design and data interpretation substantially weaken the impact of the findings.

Major Concerns:

Overexpression and Data Quality: Several critical experiments rely on overexpression systems or suffer from low data quality. This undermines confidence in the conclusions drawn from these experiments.

Concern 2-1. The reviewer provides an overall concern about overexpression, and other concerns about data quality, which are described in detail in the below concerns. Thus our responses to the below specific concerns address this overall concern.

Reliance on DNA Repair Scars: The entire study is based on analysis of repair scars, without accounting for error-free repair outcomes that leave no detectable scars. This limits the interpretability of the conclusions regarding pathway utilization.

Concern 2-2. The reviewer is concerned that we should describe the limitation of studying end joining repair that cause genetic changes (i.e., repair scars). Namely, that error-free repair cannot be measured. In response, we have described this limitation in the concluding paragraphs in the Discussion.

Oversimplification of DNA-PKcs Function: DNA-PKcs is not solely a C-NHEJ factor; it also promotes end resection through phosphorylation of CtIP, thereby facilitating HDR, SSA, and possibly MMEJ and shieldin-mediated NHEJ (Deshpande et al., Science Advances, 2020). This multifunctionality may explain some of the results shown in Figure 9 but is not adequately considered in the interpretation.

Concern 2-3. The reviewer is concerned that we have not adequately described the literature surrounding the influence of DNA-PKcs on DSB end resection, particularly the Deshpande et al 2020 paper. In response we expanded the description of the Deshpande paper in the Discussion, as follows:

“Indeed, these findings with HDR are consistent with reports that DNA-PKcs bound to DNA ends are a signal for initiation of DSB end resection via the MRE11 complex and CtIP (Deshpande et al. 2020). Specifically, addition of DNA-PK (DNA-PKcs and KU) caused activation of nuclease activity of the MRE11 complex along with CtIP that was phosphorylated during purification (Deshpande et al. 2020). Furthermore, this nuclease activity was enhanced with inhibition of DNA-PKcs kinase activity that stabilizes its binding to DNA ends (Deshpande et al. 2020).”

The reviewer asserts that Deshpande et al. 2020 shows that DNA-PKcs phosphorylates CtIP to promote its nuclease activity, which implies that DNA-PKcs kinase activity promotes end resection. While this hypothesis is a possibility, I have confirmed with the senior author of Deshpande et al. (Dr. Tanya Paull) that there is no evidence in Deshpande et al 2020 that DNA-PKcs phosphorylates CtIP. Rather, CtIP is likely phosphorylated by CDKs during purification in insect cells. Furthermore, Deshpande et al. 2020 shows that inhibition of DNA-PKcs kinase activity activates end resection, which is the opposite of DNA-PKcs kinase activity promoting end resection. As such, we have expanded the description of Deshpande et al. 2020, as described above.

Incomplete Representation of 53BP1 Pathway: 53BP1 has multiple downstream effectors—DYNLL1, RIF1, shieldin, CST, and ASTE1—each potentially leading to distinct repair outcomes. The manuscript emphasizes RIF1 without providing side-by-side comparisons with 53BP1 or other effectors, yet presents RIF1 as uniquely aligned

with 53BP1 in determining repair outcome. This oversimplifies the functional landscape of 53BP1 signaling.

Concern 2-4. The reviewer is concerned that we have not adequately described the various downstream effectors of 53BP1, and overemphasized RIF1 as the central effector protein. In response, we have removed references to RIF1 as the central effector of 53BP1, e.g., removed the word “key” from “key effector” to refer to RIF1. We also added this sentence to the Introduction: “Finally, 53BP1 has other effector proteins apart from RIF1-Shieldin that could influence DSB repair, including CTC1–STN1–TEN1, ASTE1, PAXIP1/PTIP, DYNL1, and TP53 (followed by several references).”

Flawed Experimental Design (Figure 1A): The reporter design in Figure 1A fails to account for in-frame deletions, which could occur with up to 33% probability and still yield a fluorescent signal. The authors need to demonstrate that even single amino acid deletions at the cut site eliminate fluorescence, otherwise the assay may overestimate EJ efficiency.

Concern 2-5. The reviewer raises a concern about the design of the EJ7-GFP reporter, specifically raising the possibility that in-frame deletions could also restore the GFP+ cassette. We apologize for not providing a comprehensive description and the validation of this assay, first described in Bhargava et al, 2018 Nature Comms 9, 2484, PMID: 29950655. In response to this concern, we have added key details of the reporter assay to the Results section, and to Figure 1A, as follows:

“This No Indel EJ repair outcome restores a codon that is essential for GFP function (Gly67) (reference Zacharias and Tsien 2005), causing GFP+ expression in cells, which can be measured via flow cytometry (reference Bhargava et al. 2018). The structure of the GFP+ repair product has also been confirmed by GFP cell sorting and sequencing (reference Bhargava et al. 2018).”

As a more detailed explanation, in-frame deletions cannot restore GFP+. In this reporter, we split the GFP coding sequence at the GGC codon for glycine 67 (Gly67), which is an amino acid central to the chromophore, and hence is critical for fluorescence (David A Zacharias, Roger Y Tsien. Methods Biochem Anal. 2006:47:83-120. Molecular biology and mutation of green fluorescent protein. PMID: 16335711). Indeed, structural analysis indicates that only a glycine residue at this position can support a functional chromophore (Zacharias and Tsien reference above). Finally, the neighboring residues (e.g., Ser65, Tyr66) are also critical for fluorescence (Zacharias and Tsien reference above).

We split the Gly67 codon by inserting a 46 nucleotide (nt) spacer between the first two bases (GG) and the final base (C). Two single guide RNAs (sgRNAs), 7a and 7b, target Cas9-induced DSBs to excise the 46-nt spacer. EJ between the distal DSBs without indels restores the GGC codon, which as mentioned above is critical for GFP fluorescence. Thus, the reporter is designed to only measure precise repair of the distal DSB blunt ends.

Furthermore, while this design is based on structure/function analysis of the GFP protein, we independently confirmed the expected repair product. Namely, we confirmed the expected repair product in GFP+ sorted cells by PCR amplification and sequencing analysis (Bhargava et al, 2018 Nature Comms 9, 2484, PMID: 29950655).

High Variability in Key Data (Figures 1B and 7A): The DMSO control group in Figure 1B shows high variability, which compromises statistical conclusions. Notably, overexpression of 53BP1 increases scarless EJ regardless of DNA-PKcs status, contradicting the proposed model. Similarly, DNA-PKcs inhibition consistently decreases scarless EJ, regardless of 53BP1 expression, which again contradicts the author's interpretation.

Concern 2-6.

Due to the variability of the DMSO control group, the reviewer is concerned that the statistical analysis may be compromised. In the main figures (1B and 7A), we show the GFP+ values from the reporter assay that are normalized to transfection efficiency from several independent transfections. While there is variability between the independent replicates, the assay is nonetheless sufficient to identify significant fold-effects that are less than 2-fold. However, one way to mitigate the variability among experiments is to normalize the GFP+ values to the parallel DMSO control, since the fold-effect of M3814 treatment (vs. DMSO control) in each cell line is highly consistent. Thus, in response to this concern, we have added Supplementary Figure 1, which shows the DMSO-normalized data.

Along these lines, the reviewer is concerned that our conclusion of the DNA-PKcs inhibition data is not accurate. We apologize that our conclusion was unclear. In response, in addition to including Supplementary Figure 1, we have edited the results as follows:

“However, M3814 treatment caused a reduction in No Indel EJ that was further reduced with 53BP1 loss (1.8-fold), which was complemented with expression of 53BP1 (3-fold). Namely, the fold-effect of M3814 was greater in 53BP1-KO cells vs. the parental cell line and the 53BP1-KO complemented with 53BP1 (Supplementary Figure 1). As a control, we found that 53BP1 loss does not affect M3814 inhibition of DNA-PKcs kinase activity (Supplementary Figure 2). These findings indicate that loss of 53BP1 magnifies the effect of DNA-PKcs inhibition on No Indel EJ.”

And

“Namely, the fold-effect of M3814 to inhibit No Indel EJ is magnified in RIF1-KO vs. parental and RIF-KO complemented with RIF1 (Supplementary Figure 1).”

Inconsistent Interpretation of DNA-PKcs and 53BP1 Interaction (Figure 2B): Inhibition of DNA-PKcs reduces insertion and scarless EJ while increasing deletions. Loss of 53BP1 exacerbates this shift toward deletions under DNA-PKcs inhibition. This pattern suggests additive effects, contradicting the conclusion that 53BP1 and DNA-PKcs act in the same pathway.

Concern 2-7. Similar to above concern 2-6, the reviewer is concerned that our conclusions about the genetic interactions between DNA-PKcs and 53BP1 are contradictory. We again apologize that our conclusions were unclear. In response we have edited the Results as follows:

“Altogether, these findings indicate that 53BP1 loss and DNA-PKcs disruption cause a similar increase in microhomology deletions, and that combined disruption of both factors is not additive. We suggest that DNA-PKcs and 53BP1 function in the same pathway to affect deletion size / microhomology patterns, which is a distinct genetic relationship we observed for the frequency of blunt DSB EJ (53BP1 appears to be a backup factor for DNA-PKcs). Accordingly, the genetic relationship between 53BP1 and DNA-PKcs is distinct for different aspects of DSB repair.”

Confounded Overexpression Data (Figure 9B): Overexpression results are difficult to interpret due to variable transfection efficiency and increased protein levels. The data nonetheless suggest that DNA-PKcs promotes HR in 53BP1-KO cells—53BP1-KO increases HR by 4.1-fold, while combined 53BP1 and DNA-PKcs KO only results in a 1.6-fold increase—again challenging the authors’ conclusion that these proteins act antagonistically in HDR suppression.

Most figures are well labeled and clearly presented, with the exception of Figures 1B and 7A. However, the underlying experimental design flaws and overly reductive data interpretation limit the study’s rigor.

Concern 2-8. There are two concerns.

- 1) There is a concern that transient complementation experiments have limitations. In response, we have acknowledged this limitation to the last paragraph of the Discussion, as follows:
 - a. “The genetic complementation assays involve transient expression, and the limitation of this approach is that complementation levels can vary between cells and often do not precisely match the endogenous levels.”
- 2) There is concern that our conclusion of the 53BP1 and DNA-PKcs HDR data is overly reductive and should be clearer. In particular, that the different effects of DNA-PKcs loss (PRKDC-KO) vs. kinase inhibition (M3814 treatment) should be clearly described. We responded in two ways
 - a. We have edited the Results as follows:

“Thus, 53BP1 inhibits HDR in both DNA-PKcs proficient and deficient cells, although in the latter, the fold effect of 53BP1 loss on HDR is diminished (Figure 9B). Notably, the HDR frequency for 53BP1-KO/PRKDC-KO, while higher than PRKDC-KO, was lower than 53BP1-KO. Thus, without 53BP1, loss of DNA-PKcs appears to cause a decrease in HDR. These findings underscore the difference between DNA-PKcs loss vs. inhibition (reference).
 - b. We edited the model Figure 10 for the HDR section to show that “DNA-PKcs kinase activity” inhibits HDR, to distinguish this finding vs. DNA-PKcs loss.

Reviewer #3 (Remarks to the Author):

Reviewer 3-1. We thank the reviewer for their contributions to the evaluation of the manuscript.

Reviewer #4 (Remarks to the Author):

The authors of Makins et al investigated the roles of 53BP1/RIF1 and DNA-PKcs kinase activity on C-NHEJ using GFP reporter (No indel EJ) and locus-specific large scale deletion (MA-del). They found that although 53BP1 or RIF1 loss alone do not impact end joining, they have combined impact with DNA-PKcs kinase inhibition or deficiency or with XLF. An increase in microhomology deletions is also associated with repair perturbation. However, the cooperative effect does not extend into suppressing homology-directed repair and radiosensitivity as combined perturbations are further affected beyond single target perturbations. The work could be significantly improved by addressing the below comments.

Major Comments

Although the use of HEK293 cells to generate knockout cell lines is reasonable, there is some concern over the RIF1 deficiency phenotype, which was generated using only one gRNA instead of 2 gRNAs used for generating 53BP1 deficiency and is a standard for generating permanent deficiencies as a single gRNA could lead to codon in-frame reversion mutations and confound repair pattern significance. This is a well documented phenomenon for gene therapies prior to using Cas9 and is also why a second clone is often used to discern phenotypic variation, which this reviewer notes is lacking for any deficiency in this study. The speculation about the disparity between RIF1 and 53BP1 with regard to assay differences seems unnecessary, counterproductive to supporting the MA-del assay for future use, and potentially misleading for additional downstream experiments (e.g. qPCR). To resolve this central issue, further clones should be derived for RIF1 deficiency using a 2 gRNA setup and tested to confirm whether RIF1 deficiency has a true distinction with 53BP1 at the level of DSB repair.

Concern 4-1. There is a concern that we examined only one approach to disruption of RIF1 (generating a RIF1-KO cell line with one sgRNA), particularly given discrepancies with the RIF1-KO and 53BP1-KO cell lines.

The central discrepancy was that the RIF1-KO cell line did not show an increase in HDR as was found with 53BP1-KO. Furthermore, RIF1 expression in the RIF1-KO cells caused a decrease in HDR, which indicates that indeed RIF1 likely suppresses HDR. To address these discrepancies, our response to this concern was to disrupt RIF1 using an entirely different method: transient depletion using siRNA (siRIF1).

From these experiments, we found that RIF1 depletion caused an increase in HDR (new Figure 9D, Supplementary Figure 5). Furthermore, we found that RIF1 depletion via siRNA also had similar effects on the EJ7-GFP assay as RIF1-KO (new Figure 9D, Supplementary Figure 5). In summary, using siRNA, 1) we have found that RIF1 is important to inhibit HDR in both M3814 treated and untreated cells, and 2) that RIF1 is specifically important for No Indel EJ in M3814 treated cells. These findings with RIF1 are very similar to 53BP1. We thank the reviewer for recommending that we test another way of disrupting RIF1, which has strengthened the rigor of our study.

Regarding the MA-del assay per se, RIF1-KO and 53BP1-KO were similar. Based on these reviewer's comments, we realized that we had noted a difference in No Indel EJ frequencies, which upon further experiments appears likely due to limitations of the MA-del assay vs. the EJ7-GFP assay (see Figure 5, and text at the end of page 12). So, as another response to this concern, we have clarified our conclusions with MA-del assay in the Results and Discussion.

Much of the results section go into the finer details of which sized deletion is enhanced, decreased or modified by junction structure, however, if it is not clear which of the deletion combinations for a given deletion size is used and how that may change with repair factor perturbation. Given the prominence of the data in main figures, more elaboration on drastically affected deletions should be provided with regard to how many deletion combinations represent the affecting deletion and whether this preference changes with repair perturbation (e.g., any further distinction of diversity within a specific deletion size for DNA-PKcs, 53BP1 and XLF). Perhaps a novel pattern independent of DSB position and sequence context may stand out and could be broadly applied for one, some or all of the tested repair factor perturbations.

Concern 4-2. There is a recommendation to provide specific sequence information about prominent deletions (instead of just the deletion size and microhomology amount). In response, we have generated Supplementary Tables 3 and 4 with this analysis. Namely, this figure shows the sequences of all the microhomology deletions and the relative percentage of each type for the cell lines and treatments. We thank the reviewer for recommending adding these details, which as the reviewer suggests, could inform additional studies to define mechanisms that cause particular microhomology patterns.

The authors seemed to have missed a recent publication (PMID: 38412274) describing low dose DNA-PKcs kinase inhibition promoting translocations, which are consistently reproduced here using the qPCR assay in the manuscript. It would be important for the authors to contrast their findings with that recent publication to further substantiate the observations.

Concern 4-3. There is a concern that we did not include reference PMID: 38412274, which shows that low dose DNA-PKcs kinase inhibition promotes translocations. We sincerely apologize for the oversight, and have included the recommended reference in the Results paragraph that describes the qPCR data:

“Notably, this increase in deletion frequency via M3814 treatment is consistent with a report showing that low dose DNA-PKcs inhibition causes an increase in translocations (reference for PMID: 38412274).”

Minor Comments

Inclusion of X axis labels for like figure panels (e.g. Figure 3) would be useful to include.

Concern 4-4. There is a recommendation to add x-axis labels (i.e., “insertion size” and “deletion size”) to the amplicon sequencing figures, and we have made the requested additions.

Data is all presented as percentages and not clear how many junctions are available for analysis. The authors should list the numbers of sequence reads and junction breakdown for the major subset displayed in the figures.

Information for a sequence read repository of the raw sequencer data and access to processed data from the repository should be indicated.

Concern 4-5. There are three requests about data availability (two of which are common with Concern 1-4-2). We have responded by including the requested information, as follows:

1. Request to provide the raw numbers of reads. In response, we have added a Supplementary Table 2 showing read counts for the categories of outcomes.
2. Information on the SRA repository with the MA-del *fastq* files should be added to the “Data availability statement.” In response, we have included this information, as follows: “The *fastq* files for the MA-del assay have been deposited in the Sequence Read Archive at PRJNA1301407 and PRJNA1271093, and are publicly available as of the date of publication.”
3. Processed data files should be accessible. In response, we also have included un-analyzed raw excel files from the alignment as Supplementary materials (Supplementary Data File 1, Supplementary Data File 2).

Point-by-point responses to the reviewers' comments for the manuscript "53BP1-RIF1 and DNA-PKcs show distinct genetic interactions with diverse chromosomal break repair outcomes." NCOMMS-25-36080 at *Nature Communications*.

We thank the reviewers for their positive comments and suggestions to improve the manuscript. We have responded to each concern with edits to the text. Each reviewer comment is shown in *italics/green*, and is followed by our response.

REVIEWER COMMENTS

Reviewer 1:

Overall, I am supportive of the authors' changes except for a very minor aspect of the Introduction. I support the authors thinking about the roles of 53BP1, RIF, Shieldin, and CST-Pola because I agree with the authors that the roles of these proteins need clarification in how they participate in NHEJ versus aEJ, SSA, or HR.

Prior to clarification of the distinction of NHEJ from aEJ (primarily due to TMEJ), the use of the designation C-NHEJ was understandable. But now that the distinction of NHEJ from aEJ is clearer, the prefix C- in front of NHEJ is unnecessary. I would suggest dropping the C- in front of NHEJ, as other recent reviews in this field have done.

I realize that the authors are grouping NHEJ and aEJ into one large group in the Introduction. While I am opposed to this, I can understand that the readout of the assays in this paper fit with that thinking. However, as the authors have now noted in the revision, blunt end joining in NHEJ is very infrequent due to local addition by pol mu and pol lambda and due to local limited resection by Artemis:DNA-PKcs. So blunt EJ is a very narrow view of NHEJ, which the authors acknowledge and which the authors call C-NHEJ.

If I could trouble the authors, I would suggest that they revise the couple of early paragraphs in the Intro to point out the uncertainties about 53BP1/RIF and point out that this paper was carried out to provide some insight. This would be preferable compared to the authors' current Intro which seems to want to reorganize the entire DSBR classification simply because of the uncertainties about what 53BP1 is doing.

I commend the authors' clarifications in their Responses overall, even in cases where I do not entirely agree. The authors should recognize, as said above, that most NHEJ (what they call C-NHEJ) is not precise and results in at least a few nucleotides of loss or addition. The study of precise events may account for part of their perspective on NHEJ versus other pathways.

Concern 1-1. There are two concerns with the text.

- a. The primary concern is that the term "C-NHEJ" be replaced with the simpler "NHEJ." The reviewer also suggests that this edit is important to avoid confusion that C-NHEJ could only mean precise end joining. We have made the requested change throughout the manuscript.
- b. There is a request that the gap in knowledge in the role of 53BP1 be clearly stated early in the Introduction. In response, we have edited the topic sentence of the second paragraph to focus on the gap in knowledge addressed by our study.

Reviewer #2 (Remarks to the Author):

They have addressed all our concerns.

There are no remaining concerns for this reviewer.

Reviewer #3 (Remarks to the Author):

There are no remaining concerns for this reviewer.

Reviewer #4 (Remarks to the Author):

The authors have made substantial improvements to the manuscript that strengthen their conclusions. This reviewer believes the authors have satisfactorily addressed all of the reviewers' comments.

There are no remaining concerns for this reviewer.